

# Mercury's Sodium Exosphere: An *ab initio* Calculation to Interpret MASCS/UVVS Observations from MESSENGER

Diana Gamborino[1], Audrey Vorburger[1], and Peter Wurz[1]

[1]Space Research and Planetary Sciences, Physics Institute, University of Bern, 3012 Bern, Switzerland.

**Correspondence:** Diana Gamborino(gamborino@space.unibe.ch)

**Abstract.** The optical spectroscopy measurements of sodium in Mercury's exosphere by MESSENGER MASCS/UVVS have been interpreted before with a model employing two exospheric components of different temperatures. Here we use an updated version of the Monte Carlo (MC) exosphere model developed by *Wurz and Lammer* (2003) to calculate the Na content of the exosphere for the observation conditions. In addition, we compare our results to the ones according to Chamberlain theory. Studying several release mechanisms, we find that close to the surface thermal desorption dominates driven by a surface temperature of 594 K, whereas at higher altitudes micro-meteorite impact vaporization prevails with a characteristic energy of 0.34 eV. From the surface up to 500 km the MC model results agree with the Chamberlain model, and both agree well with the observations. At higher altitudes, the MC model using micro-meteorite impact vaporization explains the observation well. We find that the combination of thermal desorption and micro-meteorite impact vaporization reproduces the observation of the selected day quantitatively over the entire observed altitude range, with the calculations performed based on the prevailing environment and orbit parameters. These findings may help to improve our understanding of the physical conditions at Mercury's exosphere, as well as to better interpret mass-spectrometry data obtained to date and in future missions, such as BepiColombo.

## 1 Introduction

The Hermean particle environment is a complex system consisting of a surface-bounded exosphere (i.e., a collisionless atmosphere down to the planet's surface), and a magnetosphere that contains volatile and refractory species from the regolith as well as backscattered solar wind and interplanetary dust (*Killen et al.*, 2007). By the end of the 1970's Mariner 10 made the first observations of the composition of the exosphere around Mercury and found hydrogen and helium (*Broadfoot et al.*, 1976). The existence of oxygen, on the other hand, although detected by Mariner 10 (*Shemansky*, 1988) has recently been brought into question; neither has it been confirmed by recent MESSENGER observations (*Vervack et al.*, 2016) nor could it be reproduced by modeling (*Wurz et al.*, 2010). It was only during the year 1985, and further on, that many ground-based observations identified the presence of sodium in the Hermean exosphere (e.g., *Potter and Morgan* (1985); *Sprague et al.* (1998); *Schleicher et al.* (2004)). Subsequent in situ observations made by MESSENGER provided a close-up look at the Hermean exosphere for over 10 Mercury years, including observations of the sodium exosphere. These observations showed that sodium emissions are temporally and spatially variable, often enhanced near north and south poles, have a moderate north-south asymmetry (*Schleicher et al.*, 2004; *Potter et al.*, 2006), are concentrated on the day-side (*Killen et al.*, 1990, 2007; *Cassidy et al.*, 2015), and show a





dawn-dusk asymmetry (*Cassidy et al.*, 2016). Due to the significant solar radiation pressure on the Na atoms in the exosphere, which can be up to half of Mercury's surface gravitational acceleration (*Smyth*, 1986; *Ip*, 1986), the sodium exosphere exhibits many interesting effects including the formation of an extended Na corona and a Na tail-like structure (*Potter et al.*, 2007; *Wang and Ip*, 2011).

Hitherto several processes have been suggested to contribute to the sodium exosphere: thermal desorption/evaporation (TD), photon-stimulated desorption (PSD), solar wind sputtering (SP) and micro-meteorite impact vaporization (MIV) (e.g. *McGrath et al.* (1986); *Hunten et al.* (1988); *Potter and Morgan* (1997); *Madey et al.* (1998); *Yakshinskiy and Madey* (1999); *Leblanc et al.* (2003); *Wurz and Lammer* (2003); *Killen et al.* (2007)). For several decades the community has been debating on the relative contribution of these mechanisms into the Hermean exosphere, and some modeling suggests that no single source
mechanism dominates during the entire Mercury year (*Leblanc and Johnson*, 2010). Laboratory experiments on lunar silicates simulants indicate that under conditions such that thermal desorption is negligible (e.g., at the lunar surface), much of the sodium exosphere can be efficiently generated by PSD (*Yakshinskiy and Madey*, 2000).

An extensive study of a subset of observations made by the Mercury Atmospheric and Surface Composition Spectrometer (MASCS) Ultraviolet and Visible Spectrometer (UVVS), on MESSENGER, was reported by *Cassidy et al.* (2015). From the
measured Na emission in the exosphere, they derived the transversal column densities (TCD) profiles, which we use in this paper's analysis. Using the Chamberlain model they interpreted the observed TCDs with two thermal components: at low altitudes, a thermal component of 1,200 K - which they suggest is due to PSD, and a hotter component at 5000 K - which they associate to MIV.

In contrast, we investigate all possible explanations using a different method. We use a Monte Carlo model in which we
use different energy distributions for the particles released from the surface according to their release mechanism. Then we calculate the exospheric particle population by describing the motion of particles under the effect of a gravitational potential and the radiation pressure from the Sun. We find that the Na observation can be explained by two combined processes: a low-energy process, TD, that dominates at low altitudes and is driven by the high surface temperature, and a comparably high-energy process, MIV, that is responsible for the Na observed at high altitudes.

We also implemented the Chamberlain model as an attempt to reproduce *Cassidy et al.* (2015) results, as well as to compare them with our MC results. The main purpose of this comparison is to examine the implications and limits of the different models in interpreting observations.

This paper is structured as follows: in Section 2 we briefly describe the previously published UVVS/MASCS observations of Na TCD that we use in this work. In Section 3 we describe the Chamberlain model, our MC model, and the modeled release
processes. The resulting density profiles and a discussion of the limitations of the models are presented in Section 5, followed by a summary and conclusions in Section 6.





## 2 Observations

In this work we use the derived data reported by *Cassidy et al.* (2015), specifically the line-of-sight column density shown in Figure 7 in their work. They derived these data from MESSENGER MASCS/UVVS observations of the Na D1 and D2 lines taken above the subsolar point on 23 April, 2012. They do so by converting the UVVS emission radiance to line-of-sight column density $N[\text{cm}^{-2}]$ using the formula $N = 10^9 4\pi I/g$, where $4\pi I$ is the radiance in kR and $g$ is the rate at which sodium atoms scatter solar photons in the D1 and D2 lines.

*Cassidy et al.* (2015) analyzed the UVVS limb scan data by fitting the Chamberlain model (*Chamberlain*, 1963) to estimate the temperature and density of the near-surface exosphere, including the effects of radiation acceleration and photon scattering. They concluded that the observations can be modeled by two thermal populations, where the bulk of the exosphere close to the surface is about 1200 K, which they attributed to PSD, and at altitudes above 500 km the temperature is 5000 K, which they attributed to MIV, both much warmer than Mercury's surface. The authors concluded that none or little evidence of thermal desorption of sodium was found. This finding was surprising and was attributed to a higher binding energy of the weathered surface that would suppress thermal desorption. They also reported that observations show spatial and temporal variation but almost no year-to-year variation, and they do not observe the episodic variability reported by ground-based observers (e.g., *Potter et al.* (2007); *Leblanc and Johnson* (2010)).

We chose these data because the observation geometry of MESSENGER MASCS/UVVS during that day, as illustrated in Fig. 3, is easy to understand and to reproduce by our model. The goal of this work is to show an interpretation from first principles of the observed line-of-sight column density of a simple case observation with a model that accounts for several release processes rather than rely solely on the Chamberlain model that accounts only for thermal release. We will consider a larger data set for forthcoming work.

## 3 Monte Carlo model description

Previously published models of the Hermean exosphere fall either into analytical or numerical models. The analytical exospheric models commonly used are based on the Chamberlain model (*Chamberlain*, 1963). However, this model restricts its applicability to spherically symmetrical distributions of gas in thermodynamic equilibrium. On the other hand, amid the numerical models, the Monte Carlo (MC) ones have become a leading method for modeling gas at collisionless regimes perfectly suited for Mercury with its surface bounded exosphere.

Here we use an updated version of the MC model developed by *Wurz and Lammer* (2003). This model represents the exosphere by a large number of model particles, typically of the order of $10^6$. We calculate the orbits of each model particle given an initial energy and angle selected randomly from a previously specified Maxwellian velocity distribution function for model particles released via TD and MIV, and non-Maxwellian ones for model particles released via PSD and SP (*Gamborino and Wurz* (2018); *Wurz and Lammer* (2003); *Vorburger et al.* (2015)). Because the gas is in a non-collisional regime we can simulate independently each release mechanism. Then we calculate each model particle trajectory under the effect of a gravitational potential and the effect of radiation pressure from the Sun. In this sense, our calculation is "*ab initio*" because we





describe the motion of the released particles using the fundamental laws of Orbital Mechanics, and we also calculate the flux of particles released from the surface for each release process from the physical conditions of the release processes.

We include the latest value for the atomic fraction of sodium in fr the surface derived from MESSENGER observations (*Peplowski et al.*, 2015) and the effect of radiation pressure on Na atoms the same way as implemented by *Bishop and Chamberlain*
5 (1989).

In the following we briefly describe the release and loss mechanisms, we explain the different assumptions concerning the Na on the surface that are important for the simulation and information about the model implementation.

## 3.1 Overview of release and loss processes

Up to now, various mechanisms have been proposed to be responsible for the input and loss of atomic species to and from
planetary exospheres (*Wurz and Lammer*, 2003; *Killen et al.*, 2007; *Wurz et al.*, 2007). Here we describe TD, solar SP, PSD, and MIV.

Each release mechanism is described by a probability energy and angle distribution function that defines an ensemble of particles with a characteristic energy from which we determine the released flux from the surface. Here we provide a brief description of the release and loss mechanisms, the mathematical expressions for the different probability distribution functions
we assume, characteristic energy, and release flux to be used in following sections.

### 3.1.1 Thermal desorption

To simulate TD we consider a Maxwellian distribution function with a characteristic energy given by the thermal energy, $E = k_B T_S$, where $k_B$ is the Boltzmann constant and $T_S$ is the temperature of the surface. The thermal speed of particles released via TD is given by the mean speed of the ensemble: $v_{\text{the}} = \langle v \rangle = \sqrt{\frac{8 k_B T_S}{\pi m}}$. Finally, the released flux is given by:

$$\Phi_{\text{TD}} = n_0 v_{\text{the}},\tag{1}$$

where $n_0$ is the number density of the species in the exosphere at the surface in units of particles per cubic meter.

### 3.1.2 Micro-meteorite impact vaporization

We determine the contribution to the exosphere by MIV in the same fashion as done by *Wurz et al.* (2010). First, we assume that Mercury's mass accretion rate for its apocentre and pericentre is, respectively, 10.7–23.0 tons/day (*Mueller et al.*, 2002).
Similarly, *Cintala* (1992) reported that the meteoritic infall on Mercury is $1.402 \times 10^{-16}$ g cm$^{-2}$s$^{-1}$ for meteorites with mass of <0.1 g, which corresponds to a radius of <0.02 m. This corresponds to a flux of 0.221 kg/s or 18.2 tons/day integrated over Mercury's surface. In contrast, *Borin et al.* (2009) reported an infall of $2.382 \times 10^{-14}$ g cm$^{-2}$ s$^{-1}$, i.e., corresponding to 1540 tons/day, which is a factor 80 times higher compared to the value by *Mueller et al.* (2002).

To calculate the exospheric densities and height profiles we derive the volatilization of surface material from the mass influx
calculated before. For our simulation of particles released via MIV we considered an average temperature of 4000 K of the





impact plume (value obtained from the range given by *Eichhorn* (1978a)). For the total number of released sodium atoms we find, accordingly, a mean MIV released flux of $2.44 \times 10^{11}$ m$^{-2}$ s$^{-1}$. We consider a uniform MIV flux over the whole planet.

### 3.1.3 Sputtering

This process refers to the impact of solar wind ions onto the surface causing the release of volatiles and refractory elements mostly near the cursp regions. The predicted value of solar wind flux to the surface near the southern cusp is four times larger than in the north (*Winslow et al.*, 2012) because of the offset of Mercury's magnetic dipole of 0.2 $R_M$ ($\sim 400$ km) northward from the planetary center (*Anderson et al.*, 2011). The northern cusp is clearly evident during the Interplanetary Magnetic Field (IMF) conditions and exhibits 40% higher plasma pressures on average during anti-sunward conditions, indicating that the effect of the IMF B$_x$ direction is present (*Winslow et al.*, 2012). However, there is a small neutral component of the solar

wind (NSW) (*Collier et al.*, 2001), which is not deflected by the Hermean magnetic field, thus permanently contributing to sputtering on the entire day-side of the planet. The energy distribution adapted with an energy cut-off (*Wurz and Lammer*, 2003) given by the binary collision limit of particles sputtered from a solid, $f(E_e)$, with energy $E_e$ of the sputtered particle, has been given as:

$$f(E_e) = C \frac{E_e}{(E_e + E_b)^3} \left\{ 1 - \left[ \frac{(E_e + E_b)}{E_{\max}} \right]^{1/2} \right\}, \tag{2}$$

where $C$ is the constant of normalisation, $E_b$ is the surface binding energy of the sputtered particle which we consider equal to 2.0 eV, taken from SRIM (*Ziegler et al.*, 2010), and where $E_{\max}$ is the maximal energy that can be transfered in a binary collision. The maximum of the energy distribution is at $E_{\max} = E_b/2$. This mechanism will release all species from the surface into space, reproducing more or less the local surface composition on an atomic level. The total sputtered flux from the surface, $\Phi_i$, of species $i$ is:

$$\Phi_i = \Phi_{\mathrm{SW}} \cdot Y_i = N_i(0) \langle v_i \rangle, \tag{3}$$

where $\Phi_{\mathrm{SW}}$ is the solar wind flux impinging on the surface, $Y_i$ is the total sputter yield for species $s$. From Eq. 3 we get $N_i(0)$, the exospheric density at the surface, with $\langle v_i \rangle$ the mean speed of sputtered particles. For the simulation we considered the mean values of particle flux and solar wind speed at Mercury for a solar wind dynamic pressure $P_{dyn} = 20$ nPa determined by *Massetti et al.* (2003): $\Phi_{sw} = 4.1 \times 10^{12}$ m$^{-2}$ s$^{-1}$ and $v_{sw} = 440$ km/s.

### 3.1.4 Photon-Stimulated Desorption (PSD)

When a surface is bombarded by photons of sufficient energy it can lead to the desorption of neutral atoms or ions. Solar photons with energy $\geq 5$ eV ($\leq \lambda = 2500$ Å) have enough energy to release sodium from the surface of regolith grains (*Yakshinskiy and Madey*, 1999). In particular, the experimental results by *Yakshinskiy and Madey* (2000, 2004) on lunar samples show that released neutral Na atoms by Electron-Stimulated Desorption (ESD) and PSD have supra-thermal speeds. Since then, several

energy or velocity distributions have been used to describe particles released via PSD, varying between Maxwellian (e.g.,



*Leblanc et al.* (2003)) to non-Maxwell distributions (e.g., *Schmidt* (2013)). However, the use of a non-Maxwellian distribution, in particular the Weibull distribution, has recently proven to be most suitable to describe this release mechanism (*Gamborino and Wurz*, 2018). The normalized Weibull distribution allows for a wide range of shapes using only two parameters for its definition. For PSD, this function is expressed as follows:

$$
f(v, v_0, \kappa) = \kappa\, \Gamma\left(1 + \frac{1}{\kappa}\right)\left(\frac{m}{3k_B T_s}\right)^{1/2}\left[(v - v_0)\sqrt{\frac{m}{3k_B T_s}}\Gamma\left(1 + \frac{1}{\kappa}\right)\right]^{\kappa - 1} \times
$$

$$
\times \exp\left[-\left((v - v_0)\sqrt{\frac{m}{3k_B T_s}}\Gamma\left(1 + \frac{1}{\kappa}\right)\right)^{\kappa}\right] \tag{4}
$$

where $k_B$ is the Boltzmann constant, $m$ is is the mass of the species, $T_s$ is the surface temperature, $\Gamma$ is the Gamma function, $v_0$ is the offset speed, and $\kappa$ is the shape parameter of the distribution. The parameters have been derived as $\kappa = 1.7$ and speed offset of $v_0 = 575$ m/s by *Gamborino and Wurz* (2018).

## 3.2 Loss processes

We include the following exospheric loss processes: (1) gravitational escape, (2) surface adsorption, and (3) ionization. The escape speed from Mercury's surface is $v_{\mathrm{esc}} = 4.3$ km/s, which is small enough to allow the escape of many exospheric particles, particularly the light ones. We compute the fraction of atoms lost by photo-ionization at each time step in the trajectory calculation, and we use the typical photo-ionization rates of Na at Mercury, which are $7.2 \times 10^{-6}$ s$^{-1}$ and $7.9 \times 10^{-6}$ s$^{-1}$ during low and high Solar activity, respectively (http://phidrates.space.swri.edu/). On average, it takes a few hours until a sodium atom is ionized, which gives enough time to complete an exospheric trajectory for most released particles. Sodium atoms will go back to Mercury's surface and be adsorbed, unless they are lost due to ionization or escape Mercury's gravity.

In Figure 1 we show the different energy spectra for Na from the probability distribution functions, each normalized to a maximum of one, for the four release mechanisms. Atoms released via TD have an energy distribution dependent on the local surface temperature, which is represented by the *solid-black* curve, corresponding to a characteristic energy of 0.06 eV and a temperature of 594 K. Atoms released via this process have a relatively low characteristic energy compared to the escape energy of sodium atoms from Mercury's surface, which is 2.07 eV for Na atoms. Thus, TD leads to a near-surface Na exosphere that does not contribute to the planetary loss. On the other hand, atoms released via SP (*dashed* curve), have significantly higher characteristic energy (1 eV) and a distribution skewed to higher energies (see Eq. 2), thus reaching higher altitudes and contributing to the planetary loss. Since ion fluxes onto the surface and sputter yields are low, sputtered atoms can only form a low-density exosphere (*Wurz et al.*, 2010). A long-tailed positively skewed distribution also describes the energy distribution for for PSD release (see Eq. 4) shown as the *dashed-dotted* curve in Fig. 1, which has a characteristic energy of ∼0.1 eV, thus particles can reach higher altitudes compared to those released via TD. As can be seen in the plot, the energies are mostly below escape. Finally, atoms released via MIV (*dotted* curve) are modeled by a thermal distribution with temperatures of 4000 K, thus higher characteristic energy (0.34 eV) compared to the regular TD. MIV contributes to the exosphere in a

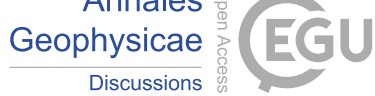



comparable amount like SP (*Wurz et al.*, 2010), but, unfortunately, the micro-meteorite influx is not known very accurately, as discussed above.

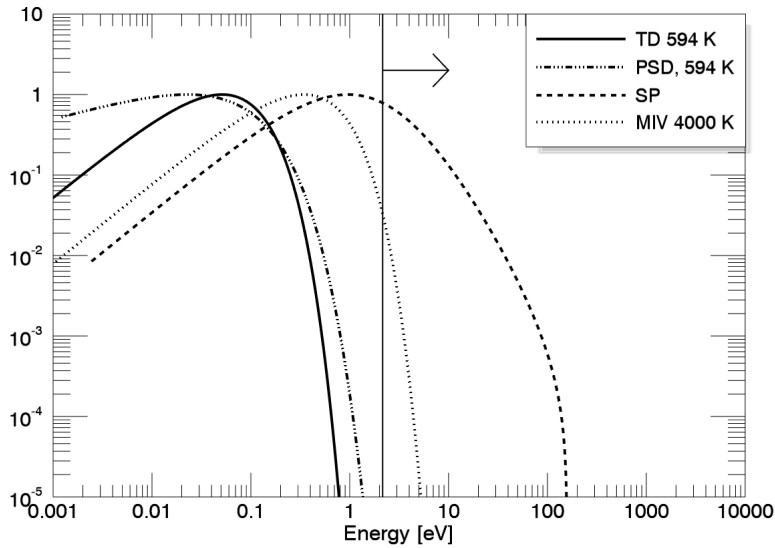

**Figure 1.** Normalized energy distribution functions for several release mechanisms active on Mercury's surface to produce the sodium exosphere. The vertical solid line marks the value at which sodium atoms have enough energy to escape the gravitational field of Mercury.

### 3.3  Sodium in and at the Surface

*Smyth and Marconi* (1995) introduced a qualitative description of the fate of sodium atoms ejected in Mercury's environment using two populations, the "source" and the "ambient" atoms. A few years later, *Morgan and Killen* (1997) expanded the description to include the diffusion of sodium from inside the regolith to the surface and the description of the expanding vapor cloud after the impact of micrometeorites. Here we consider these two populations; a first population, the so-called "source atoms", which are atoms chemically-bonded in the minerals on the surface and are released by high-energy processes, either by MIV or SP. The source atoms are predominantly ionically bonded to the oxygen in a bulk silicate (*Madey et al.*, 1998) with binding energy larger than 0.5 eV.

The released Na atoms may either escape, or fall back on the surface and become ambient particles after few impacts on the surface (measured in experiments by *Yakshinskiy and Madey* (1999, 2000, 2004)). These particles are thermally accommodated to the local surface temperature and have counterparts absorbed in the regolith with a binding energy less than 0.5 eV, according to *Hunten et al.* (1988). We model this population by low energy processes such as TD and PSD. An illustration of the different release, returning, and loss fluxes is shown in Figure 6.



## 4 MC model Implementation

Following *Wurz et al.* (2010), we model each atom trajectory by designating an initial energy and an ejection angle selected

at random from the prescribing distribution function for the appropriate ejection mechanism. Then, the particle trajectory is

determined for discrete altitude steps with start point at the surface until the particle either falls back to the surface, gets ionized

somewhere on its path and thus is lost from the neutral exosphere, or leaves Mercury's gravity field thus leaving calculation

domain (which is given by the Hill radius).

**Table 1.** Parameters for simulation.

| Orbital and geographical parameters | All processes |
|---|---|
| Orbital radius ($R_{orb}$[AU]) | 0.458 |
| True Anomaly Angle (TAA) | 160° |
| Solar Zenith Angle (SZA) | 0° † |
| Longitude | 0° |
| **Physical parameters** | |
| Number of model particles | $10^6$ |
| Surface temperature [K] | 594 |
| Surface Na atomic fraction | 0.0234* |
| Global meteoritic infall | 10.7–23.0 tons/day** |
| UV flux at surface $\phi_{ph}$ [$m^{-2}s^{-1}$] | $1.57 \times 10^{20\diamond}$ |
| Radiation acceleration ($b_{rad}$ [cm s$^{-2}$])‡ | 28 |

* Intermediate Composition (*Peplowski et al.*, 2015).

†Latitude=0°.

** *Mueller et al.* (2002)

‡*Smyth and Marconi* (1995).

◇ for PSD.

After simulating all model particles' trajectories, we compute the species' density and column density profiles as a function

of altitude and tangent altitude by applying the boundary conditions given from the particle release mechanism. For calculation

of the tangent altitude integration we assume a radially symmetric exosphere (see Figure 3). In Figure 2 we show an example

of the tangent column density profiles computed for the different release mechanisms. All processes were simulated with and

without radiation pressure to compare them. We fixed all profiles at altitude zero to show how the TCD profile would look for

different mechanisms given the same surface density.

### 4.1 Simulation parameters and geometry

To reproduce the measured TCD profile reported by *Cassidy et al.* (2015), we have to know some input values for the simulation

parameters. Using SPICE and Mercury's ephemeris data we find the corresponding parameters for the time of the observations





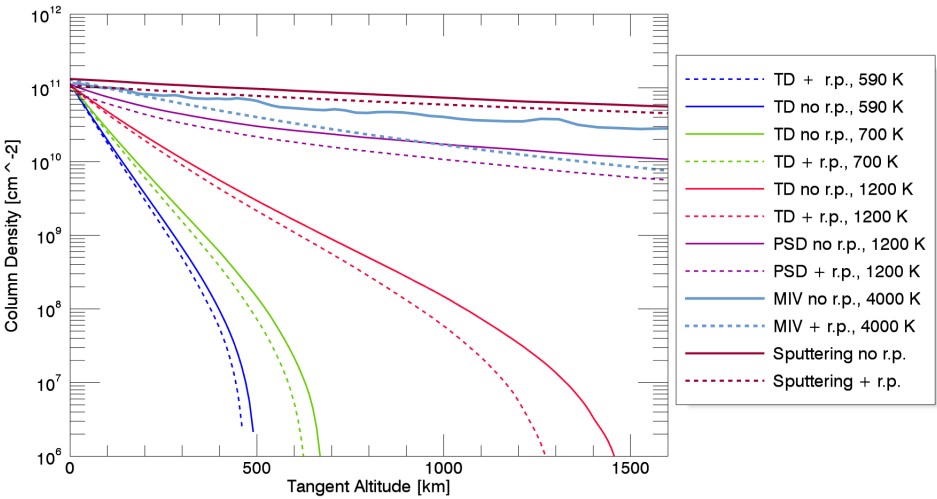

**Figure 2.** Column density profiles as a function of tangent altitude for Na for the different release mechanisms normalised to a column density of $10^{11}$ cm$^{-2}$ at the surface: TD, PSD, MIV, and SP with and without radiation pressure (r.p.) centered at subsolar point and for different surface temperature values. The simulation was done with an ensemble of $10^6$ particles, and Mercury at a TAA=160° and $R_{orb} = 0.458$ AU.

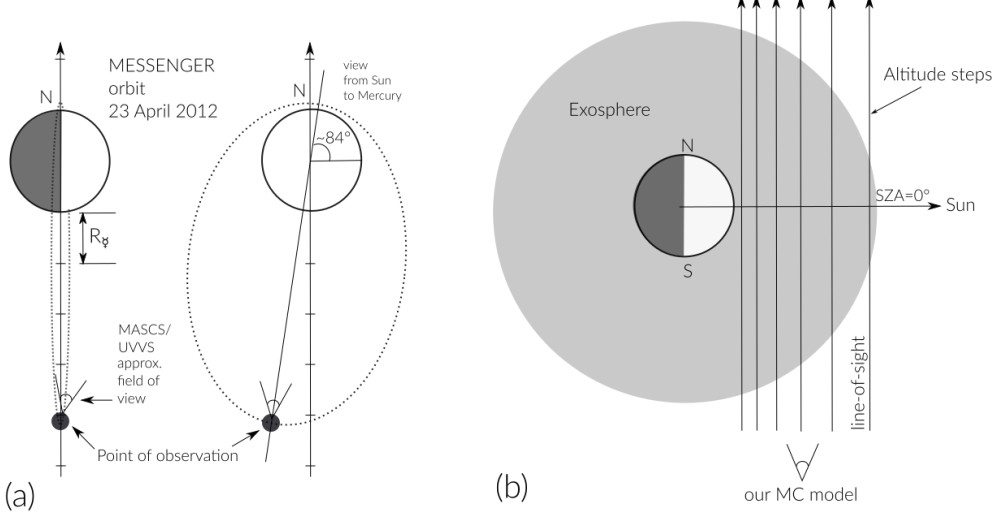

**Figure 3.** (a) Representation to scale of MESSENGER's orbit around Mercury during the time of observations (orbits parameters taken from www.nasa.gov). (b) Scheme of the line-of-sight used in the simulations (distance between altitude steps is not to scale). *Solid-black* lines on (b) represent the altitude steps for integration of the the column density when SZA=0°.

(listed in Table 1). The day-side limb-scans were taken when Mercury had a TAA of approximately 160°, MESSENGER was near the apogee, and with the UVVS line-of-sight pointing approximately northward - as shown in the illustration in Figure 3; see also top of Figure 2 in *Cassidy et al.* (2015). During the day of the observation MESSENGER made three orbits around

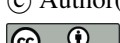



Mercury with an orientation such that its orbit plane was almost perpendicular with respect to the light rays coming from the Sun, as represented in the illustration in Figure 3 (mind that this is just a simplified representation of the real situation). Each altitude profile extends from just above the surface, as low as 10 km, to several thousand kilometers above the surface.

Given the position of the spacecraft with respect to Mercury and the direction of the UVVS line-of-sight during the time of observation, we simulate particles ejected at latitude= 0°, SZA= 0°, and calculated the contribution to the TCD from this position. This is the closest to the real observation geometry that our model can compute. In Figure 3.(a) we illustrate the orbit geometry and position of the MESSENGER spacecraft during the time of observation.

In Figure 3.(b) we also illustrate the orientation of the tangent (to the surface) altitude steps we used in our model for the integration of the column density. The dawn/dusk asymmetry is not considered because the time of the observation does not include these regions.

After computing the density profiles we integrate along the line-of-sight for different altitude steps and for SZA=0° to obtain the limb scans as a function of tangent altitude.

The radiation pressure depends on the true anomaly angle and the solar zenith angle, and we use the value for radiation acceleration taken from *Smyth and Marconi* (1995) for the given TAA during the time of observations. We assume the same orbital and physical parameters for our implementation of the Chamberlain model modified by adding radiation pressure.

### 4.1.1 Temperature model

According to the infrared measurements made on Mariner 10 (*Chase et al.*, 1976), which only include observations from the night-side up to 8:00 LT on the day-side, and considering Mercury as a blackbody emission radiator, the extrapolated day-side surface temperature as a function of latitude $\phi$ and longitude $\theta$ must follow a "1/4"–law with illumination angle, and decrease like "$1/r^2$" where $r$ is the distance of Mercury to the Sun. Neglecting the thermal inertia of Mercury's lithosphere, the local *day-side* surface temperature for $0 < |\theta| < \pi/2$ can be written as:

$$T_s(\phi,\theta) = T_{min} + (T_{max} - T_{min})(\cos\phi\cos\theta)^{1/4}$$

where $T_{max}$ is the effective temperature at the subsolar point, $T_{min}$ is the night-side temperature, the longitude $\theta$ is measured from the planet-Sun axis, and the latitude $\phi$ is measured from the planetary equator. To determine the effective temperature at subsolar point we used the blackbody Stefan-Boltzman law to obtain:

$$T_{\text{eff}} \approx T_{Sun}\left[\left(\frac{R_{Sun}}{R_{orb}}\right)^2 \frac{1-\alpha}{\epsilon}\right]^{1/4} \tag{5}$$

where $T_{\text{eff}}$ is the effective temeprature of the surface, $R_{orb}$ is the distance to the Sun, $R_{Sun}$ is the solar radius, $T_{Sun} = 5778$ K is the effective solar temperature, $\alpha = 0.07$ is the albedo (*Balogh et al.*, 2000) and $\epsilon = 0.9$ is the emissivity (*Murcray et al.*, 1970; *Saab and Shorthill*, 1972; *Hale and Hapke*, 2002). During the day of observations Mercury was at a distance of $R_{orb} = 0.458$ AU to the Sun. Using this value and the temperature formula from above (Eq. 5), we calculated a surface temperature of $T_{\text{eff}} = T_S = 594$ K at the subsolar point. This is the temperature we used for the TD and PSD calculations.



## 5 Results and Discussion

Using the parameters listed in Table 1, we simulate sodium atoms released via TD, PSD, MIV, and SP and calculated the exospheric density profiles up to $10^5$ km. The integral along the line-of-sight gives us the column density and if we choose the tangent altitude we also obtain the TCD as a function of altitude. The surface TCD and released flux obtained from the simulation for each mechanism are listed in Table 2. Independently, we also implemented the Chamberlain model in the same fashion as *Cassidy et al.* (2015) did to compare it to our results (see description in Appendix).

In Figure 4 we plot the result from our MC simulations for the sodium TCD as a function of altitude for: TD (*dashed-black* curve), MIV (*dashed-grey* curve), SP (*dash-dot-dot-dot grey* curve), and PSD (*vertical-dash black* curve). The derived TCD from the day of observation of the Na emission data is shown as the black cross marks (*Cassidy et al.*, 2015). The results using the Chamberlain model with the modification for radiation pressure is also plotted: the curve with *square* symbols corresponds to a surface temperature of 594 K, and the curve with *circle* symbols corresponds to an assumed temperature of 2500 K, with the TCD at the surface of this component adjusted to match the observations. The sum of the two Chamberlain profiles are represented by the *solid-light-grey* curve.

In Figure 5, the sum of TD and MIV profiles (*solid-black* curve),

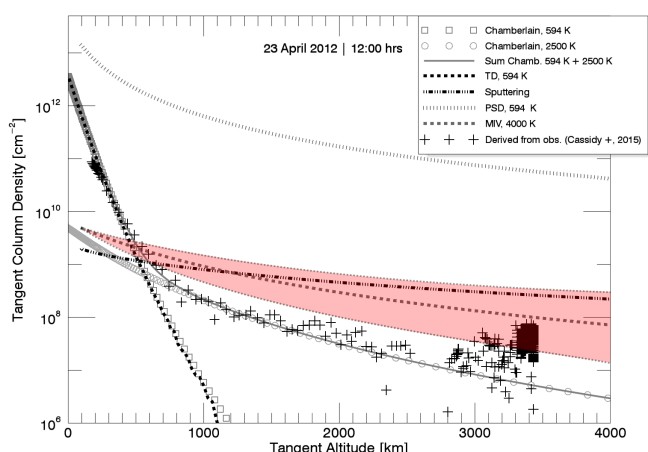

**Figure 4.** Plots of the derived TCD from observations (black crosses), of the results from our calculations with no correction factors, and of the results using the Chamberlain model. The resulting TCD profiles from our MC model are plotted as follows: TD is the *dashed-black* curve, MIV between 3000 K and 5000 K is the shaded red area, the *dashed-grey* curve is MIV with a mean temperature of 4000 K, SP is the *dash-dot-dot-dot grey* curve, and PSD is the *vertical-dash* curve. The resulting profiles using the Chamberlain model are represented as the *square-* and *circle*-symbols curves. The sum of the two Chamberlain profiles represented by the *solid-light-grey* curve.

At low altitudes the Chamberlain model and the TD simulation for the surface temperature agree very well, and both also agree reasonably well with the observations, which is expected from a Maxwellian population thermalized with the surface. Note that the calculated thermal profiles, both with the Chamberlain model and with our MC model, are based on the surface temperature, which was derived separately from the orbit position of Mercury, and were not fitted to the data. The fall off





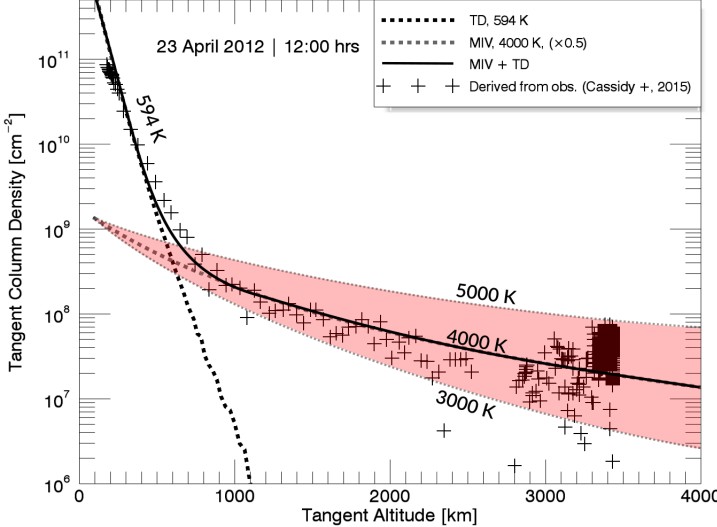

**Figure 5.** Plots of the derived TCD from observations (black crosses), and of the results from our calculations: the *dashed-black* curve for TD, the results for MIV between 3000 K and 5000 K is the shaded red area, the *dashed-grey* curve is MIV with a mean temperature of 4000 K*dashed-grey*. MIV curves were multiplied by 1/2. The sum of TD and MIV profiles is shown as the *solid-black* curve.

above 1000 km of the TD 594 K curve is a result of the loss of neutral sodium atoms due to ionization. Compared to any other mechanism modeled here, TD is the best match to the observations at low altitudes.

At altitudes above $\sim$ 600 km the Chamberlain model for 594 K shows too low densities and only a high-temperature component of 2500 K fits the data, similar to *Cassidy et al.* (2015). The SP curve is too flat and does not match the observations
at any given altitude. As described further on, and as shown in Figure 7, the sputerred component of the tangent column density is expected to be substantial mainly at high latitudes, but the observation geometry gives preference to mechanisms happening at low latitudes.

The limitation of Chamberlain theory (*Chamberlain*, 1963) in this context is that it was originally developed for Earth's exosphere where the only controlling factors considered are the gravitational attraction and the "*thermal energy conducted*
*from below*" (also known as exobase). For the case of Mercury, the only layer below the exosphere is the surface. Meaning that, the only "source" mechanism of exospheric particles considered in Chamberlain model is what we consider in our MC model as Thermal Desorption.

Using the Chamberlain theory implies that the only way to increase the particles' characteristic energy (and thus able to reach high altitudes) is by increasing the surface temperature. One way to do this is by micrometeorite impacts, which can lead
to a Maxwellian exospheric population with a temperature of the order of a few thousand Kelvin, which can explain the high-energy component in the observations. Other way to increase the temperarure is by heating the surface with solar radiation. But as we know, the surface temperature of Mercury at the subsolar point and at TAA=160° is 594 K. This temperature is not enough to let particles reach high altitudes, much higher temperatures are needed as shown in Figure 4. Consequently, the



Chamberlain model works fine only for an exospheric population that is in thermal equilibrium with the surface temperature. For other non-Maxwellian and more energetic populations, the Chamberlain model is inadequate.

Hence, it is inevitable to consider other, non-thermal and more energetic, release processes to explain the Na exosphere at higher altitudes. Our results show that PSD and MIV are two possible non-thermal and high-energy mechanisms that can
explain the observations at high altitudes, as shown in Figure 4. Our MC model of the MIV TCD profile gives a good match with the observed data at high altitudes with a temperature of 4000 K$\pm$1000 K, consistent with *Eichhorn* (1978a). An even better match is reached by adjusting the surface column density by only a factor of 0.5. On the other hand, the PSD TCD profile does not fit quite as well to the obervations and it has to be scaled with a factor of $4 \times 10^{-4}$ to match part of the tail, which will be addressed below.Uzcanga
Since TD is competing with PSD for the ambient Na atoms on the surface, the Na available for PSD is much less, or none available at all, in the extreme case (as explained below). Therefore, the plotted curve of PSD is just an upper limit. The TCD profile for SP is also plotted in Figure 4 for precipitating ion fluxes of the cusp region. Since the observations are at the subsolar point, the surface is shielded from precipitating ions (*Kallio and Janhunen*, 2003) and the SP contribution to the exosphere for these observations has to be considered as an upper limit as well. Moreover, the TCD from SP falls off much less with altitude
than the observations.

### 5.1 Source and loss fluxes

The sodium loss from the exosphere has to be supplied by Na from the surface to sustain a stable exosphere over several Mercury years as it was observed (*Cassidy et al.*, 2015). As mentioned in Section 3.3, the source population of Na to the exosphere is considered to come mainly from the release via SP and MIV. The fraction of this population that comes back
to the surface will become the ambient population and be available for TD and PSD. The conservation of mass allows us to quantify the amount of Na in the ambient population at the subsolar point, which is available for TD, by considering that the sum of the Na diffused from regolith to the surface plus the return fluxes from MIV and SP has to be equal the loss due to TD. Mathematically, the latter is stated as:

$$\Phi_{\text{source}}^{\text{Ambient}} = \Phi_{\text{Diff.}} + \Phi_{\text{return}}^{MIV} + \Phi_{\text{return}}^{SP} \qquad (6)$$


$$\Phi_{\text{loss}}^{\text{Ambient}} = \Phi_{\text{release}}^{TD} \cdot \chi_{\text{Tot}}^{TD} \qquad (7)$$

where $\chi$ is the total fraction of Na loss, which includes the losses by ionization and gravitational escape. $\Phi_{\text{Diff.}}$ is the diffusion-limited exospheric flux calculated by *Killen et al.* (2004) to be $< 10^7$ cm$^{-2}$ s$^{-1}$. Note that the fluxes' subscripts *source* and *loss* are calculated just for the ambient population and does not represent the global flux. For a steady state system:

$$\Phi_{\text{source}}^{\text{Ambient}} - \Phi_{\text{loss}}^{\text{Ambient}} = 0 \qquad (8)$$





Combining Equations 6, 7, and 8 we derive $\Phi_{\text{release}}^{TD}$ as follows:

$$\Phi_{\text{release}}^{TD} = \frac{\Phi_{\text{Diff.}} + \Phi_{\text{return}}^{MIV} + \Phi_{\text{return}}^{SP}}{\chi_{\text{Tot}}^{TD}} = \frac{1.0 \times 10^7 \text{cm}^{-2}\text{s}^{-1} + (5.4 \times 10^5 + 3.28 \times 10^5)\text{cm}^{-2}\text{s}^{-1}}{0.0102} = 1.06 \times 10^9 \text{cm}^{-2}\text{s}^{-1} \quad (9)$$

From Eq. 9 we can derive the surface density, and radial column density, as $n_0 = \frac{\Phi_{\text{release}}^{TD}}{v_{th}} = 1.44 \times 10^{10}\text{m}^{-3}$. The radial column density, $NC$, can be approximated as: $NC \approx n_0 \cdot H = 8.21 \times 10^{14}$ m$^{-2} = 8.21 \times 10^{10}$ cm$^{-2}$. $v_{th} = \sqrt{8k_B T/\pi m} = 739$ m/s

is the thermal speed. The rest of the results of our simulations, together with the derived quantities for the ambient population are shown in Table 2.

**Table 2.** Results from the MC simulation.

| Property | TD | MIV | PSD | SP |
|---|---|---|---|---|
| Scale height [km] | 57 | 431 | 232 | 748 |
| Surface density ($n_0$ [m$^{-3}$]) ‡ | $1.28 \times 10^9$ | $5.97 \times 10^6$ | $3.45 \times 10^{10}$ | $3.72 \times 10^6$ |
| Column density (NC [cm$^{-2}$]) | $7.29 \times 10^9$ | $2.57 \times 10^{12}$ | $8.0 \times 10^{15}$ | $2.79 \times 10^{12}$ |
| Total released Na flux [cm$^{-2}$ s$^{-1}$] | $9.51 \times 10^7$ | $5.74 \times 10^5$ | $3.63 \times 10^9$ * | $1.32 \times 10^6$ |
| Total fraction of lost particles[†] | 0.0102 | 0.0589 | 0.0307 | 0.7510 |

[†]Including the losses by gravitational escape and ionization.

[‡]Based on Na surface fraction from Intermediate Composition (*Peplowski et al.*, 2015), and used for MIV, PSD, and SP.

* Adjusted to observations by a multiplication factor of $4 \times 10^{-4}$, and is considered an upper limit.

Figure 6 is a scheme illustrating the Na release processes and fate due to the different release mechanisms. For the given observations, we calculated the Na release flux from the surface and determined the losses for each process. The fraction of Na that is not lost returns to the surface and becomes part of the ambient population available for TD and PSD.

Figure 7 is an illustration of the spatial distribution of the derived released Na fluxes as a function of solar zenith angle. The radial scale repesents the magnitude difference of the release flux "intensity", $I(\alpha)$, for the different release mechanisms (the units are arbitrary). Note that it does not represent the spatial distribution of Na depending on the mechanism. At the subsolar point, the main contribution of Na to the exosphere is TD with a release flux of two orders of magnitude higher compared to the other release processes, and with an expontential decay towards the poles because of the temperature dependece of

sublimation. The solar wind sputtering on the surface acts mainly at high latitudes ($\alpha \approx \pm\pi/2$), and is also a strong funtion of latitude decay. We consider that MIV acts uniformily on the entire planet, thus the release flux is not angle dependent. PSD has a cosine dependence with SZA, but since it competes with TD it is most important at mid- to high latitudes.

This illustration shows the main release mechanisms given a certain line-of-sight observation geometry. For instance, the ground based observations done by *Schleicher et al.* (2004) during Mercury's transit had a limb line-of-sight similar to the

horizontal line shown in this figure. The main contribution for those limb observations Sun. Neglecting the thermal inertia of Mercury's lithosphere, the local when $\alpha \approx \pm\pi/2$ include a sum of SP, which provides a sputter Na exosphere and ambient Na, and PSD from the ambient population, which was already discussed by *Mura et al.* (2009). On the other hand, the vertical line-of-sight path crossing TD and MIV represents the line-of-sight of the approximate direction of MESSENGER's field of view during observations on 23 April, 2012 (*Cassidy et al.*, 2015) and discussed here.





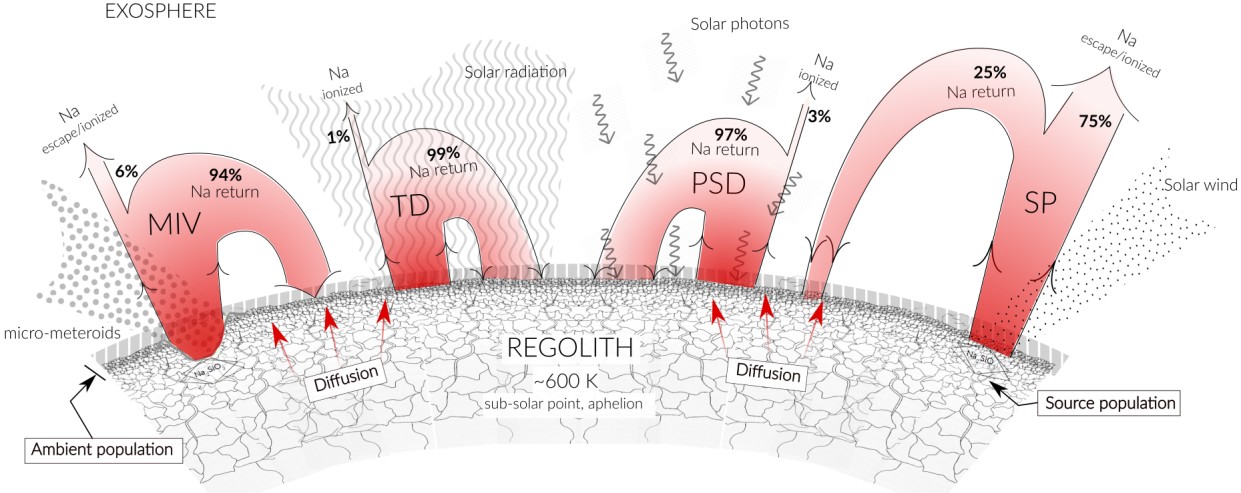

**Figure 6.** Scheme illustrating the different released fluxes of Na due to the different release mechanisms from the mineral compound (the source population) and from surface (the ambient population). The illustration is not to scale.

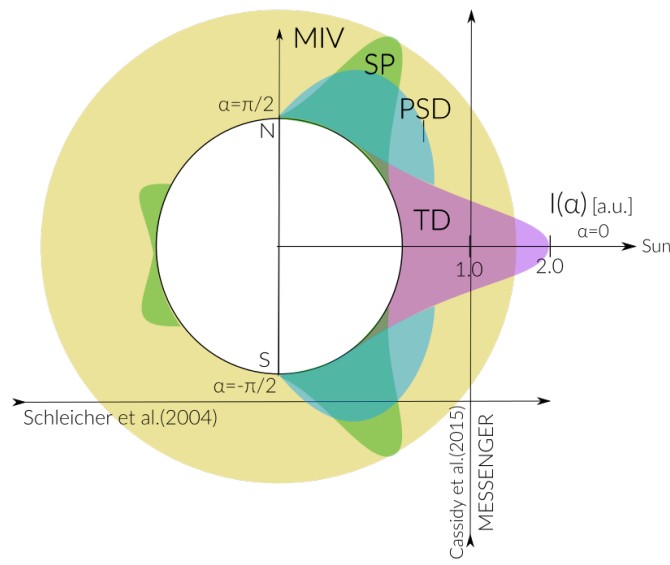

**Figure 7.** Scheme of the intensity distributions for the different release mechanisms as a function of solar zenith angle. The units are arbitrary but orders of magnitude are taken from the results from our model. The horizontal arrow is an example of the line-of-sight of the ground-based observations by *Schleicher et al.* (2004). Vertical arrow represents the approximate direction of MESSENGER's field of view during observations on 23 April, 2012 (*Cassidy et al.*, 2015).





## 5.2 Is there a permanent Na atom layer on the surface?

Reviewed earlier reports (e.g. *Leblanc et al.* (2003)) conclude there is a non-uniform spatially distributed but permanent "reservoir" of atomic Na available on the surface, forming part of the ambient population. This reservoir is not strongly chemically bounded to the mineral grains, but it is physisorbed on the surface and being on the surface it is also not part of the exosphere.

Let us consider sodium adsorbed in an atomic state on the surface rather than chemically bounded to the crystal structure in the regolith (consistent with laboratory experiments by *Yakshinskiy and Madey* (2000, 2004)). We can calculate the theoretical Na gas density above the surface and the evaporated flux by using the empirical equation for the Na vapor pressure (*Lide*, 2003) and considering a surface temperature of $T = 594$ K at subsolar point, this gives a sodium vapor pressure of $P_0 = 7.44$ Pa.

The corresponding theoretical evaporated flux of sodium atoms from this surface reservoir would be $6.71 \times 10^{21}$ atoms per $cm^{-2} s^{-1}$.
This value is 14 orders of magnitude higher than the thermal Na release flux we derived from measurements, which is $4.34 \times 10^{8}$ atoms $cm^{-2} s^{-1}$. Even if the atomic Na is bound to the surface with a higher binding energy, the sublimation flux will still be enormous at this temperature Thus, it is clear that the TD flux is limited by the availability of ambient Na on the surface, and given the large difference in theoretical and derived release fluxes, all the ambient Na is in the exosphere and not on the surface near the subsolar point. This implies that all the ambient sodium released from the bulk or that has fallen back onto the surface
near the subsolar point will be immediately evaporated to the exosphere leaving no atomic Na left on the sun-lit surface. Thus, based on these considerations, PSD can not compete to desorb sodium because there is no Na left available on the surface.

These interpretations follow the line of what was previously deduced from experiments by *Madey et al.* (1998) carried out with various oxide surfaces that resembled the hermean and lunar regolith. The authors found that at temperatures in the range of $\sim 400 - 800$ K, TD of fractional Na monolayers occurs for low alkali coverages, with the desorption barrier (and surface
lifetimes) increasing on a radiation damaged surface. Specially, their measurements indicate that at equatorial regions TD rapidly depletes the alkali atoms from the surface reservoir, whereas it is less efficient at high latitudes. This is consistent to our finding that at low altitudes and at the subsolar point, TD is the dominant process and is responsible for the exospheric Na at low altitudes up to about 600 km. On the other hand, MIV is a plausible mechanism to explain the high-energy component, since it releases Na present in the bulk, and thus is not limited by the availability of Na on the surface.

## 25  6   Conclusions

We present the results of our Monte Carlo model of the hermean sodium exosphere and compare them with the sodium tangent column density profile derived from MASCS/UVVS measurements during the day 23 April, 2012 (*Cassidy et al.*, 2015). Using the correct parameters for TAA and $T_S$ for the day of the observations, we simulate the density profiles of sodium atoms ejected from Mercury's surface through TD, PSD, MIV, and SP as release mechanisms.
We reproduce the derived sodium TCD profile as a function of altitude: below 500 km release via TD dominates governed by a surface temperature of 594 K corresponding to a characteristic energy of 0.06 eV. Because of the very high release fluxes, the Na in the exosphere near the surface is due to TD, limited by the supply of available Na atoms on the surface. Only at higher altitudes the contribution by MIV prevails up to the observed 4000 km with a characteristic energy of 0.34 eV.





For the first 500 km with the MC model TD results agree well Chamberlain model using a local surface temperature of 594 K, and both agree with the measurements. As we go further away from Mercury's surface though, there is a more energetic component of Na atoms in the exosphere which we find to be the result of MIV.

We have also shown that if there would be an ambient sodium layer available on the surface, this would have to be immediately evaporated due to the high volatility of Na at such a high surface temperature prevailing at the given observation time. The Na release by TD is strongly limited by the supply of free Na to the surface at the prevailing surface temperature. Moreover, release by PSD can only be responsible for the Na exosphere population at higher altitudes, because of the higher energies of the released Na atoms. However, we find that we can only give an upper limit for the release of Na via PSD for the investigated observations.

Our results diverge substantialy from the results by *Cassidy et al.* (2015). While their paper is elucidating and explains the derived observations in terms of two thermal release populations using the Chamberlain model, it seems they arrive at a confounding near-to-the-surface sodium temperature of 1,200 K. Using their assumptions and the Chamberlain model we get good agreement of our model (MC and Chamberlain) for the calculated surface temperature, which is half their value, and the results using our MC model confirms the same number.

The use of mass spectrometers is crucial to study the surface composition of Mercury and ultimately to understand the origin of species found in the exosphere since they come from the regolith and crust (except for the noble gases, hydrogen and a few volatile species such as sulphur, which are abundant in the micro-meteorite population). To prepare for the SERENA investigation (*Orsini et al.*, 2010), to be performed on board of ESA's BepiColombo planetary orbiter (*Milillo et al.*, 2010), we have updated and extended our MC model, originally developed by *Wurz and Lammer* (2003), which is a tool to quantitatively predict exospheric densities for several release processes using the actual physical parameters of the release process.

*Acknowledgements.* We gratefully thank the Swiss National Science Foundation (200020–172488) for supporting this work. The first author also wants to thank André Galli for their support and help while writing this paper.

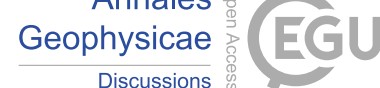



## Appendix A: Chamberlain Model Implementation

Chamberlain's model (*Chamberlain*, 1963) is based on Liouville's Theorem applied to a collisionless exosphere where the gravitational attraction and the thermal energy conducted from below are the controlling factors. The Liouville equation is solved using a Maxwellian distribution as the boundary condition at the exobase. The velocity distribution is then is the flux coming from below the regolith by diffusion integrated in the region allowed by the trajectory in a gravitational field and over the velocity space, which can be divided into different populations that represent different types of particle orbits: *ballistic* (captive particles whose orbits intersect the critical level, i.e., surface in Mercury's case), *satellite* (captive particles orbiting above the critical level), and *escaping* (particle's velocity is larger than the escape velocity). This leads to analytic expressions for the density distributions and the loss flux. The number density at a given altitude $r$ is given by:

$$n(r) = n(r_c)\exp[-(\lambda_c - \lambda)]\zeta(\lambda) \tag{A1}$$

where the parameter $\lambda$ represents the absolute value of the potential energy expressed in units of $k_B T_c$ as follows:

$$\lambda_c(r) = \frac{GMm}{k_B T_c r} = \frac{v_{esc}^2}{V^2} \tag{A2}$$

where $v_{esc} = (2GM/r_c)^{1/2}$ is the escape velocity and $V = (2k_B T_c/m)^{1/2}$ is the most probable Maxwellian velocity (thermal velocity), $G$ is the gravitational constant, $M$ is the mass of the planet, $m$ is the mass of the species, $k_B$ the Boltzmann constant, $T_c$ the exobase temperature, and $r$ the radial distance from the center of the planet. The sub-index $c$ stands for *critical level* that corresponds to the exobase, which corresponds to Mercury's surface in this case.

Equation A1 is a combination of the barometric density equation with a partition function $\zeta(\lambda)$, where $n(r_c)$ is the density at the critical level. The factor $\zeta$ may be regarded as the fraction of the isotropic Maxwellian distribution that is present at a given altitude, subject to conservation of energy and angular momentum. For no dynamical restrictions to the orbit, $\zeta = 1$, which leads to the generalized form of the (isothermal) barometric law. However, at substantial distances above the critical level the barometric law breaks down because the pressure at large distances is decidedly directional and the mean kinetic energy per atom decreases. The atmosphere is not strictly in hydrostatic equilibrium, moreover it is expanding slightly, i.e., some matter is being lost, which in the kinetic theory corresponds to evaporative loss. To treat the density distribution accurately it is necessary to examine the individual particle orbits which is the case when $\zeta \neq 0$. The analytical expressions of $\zeta$ for each class of particle orbits can be found in *Chamberlain* (1963). The effect of radiation pressure on sodium atoms was also incorporated following *Bishop and Chamberlain* (1989). Examples of sodium column density profiles for the different types of trajectories are displayed in Fig. A1 considering a surface temperature of $T_c = 594$ K.

On the other hand, sodium atoms in the atmosphere of Mercury can be accelerated by solar radiation pressure resulting from resonant scattering of solar photons. In earlier works it has been suggested that radiation pressure could sweep sodium off the planet, provided that the sodium is non-thermal [e.g. *Ip* (1986); *Bishop and Chamberlain* (1989); *Wang and Ip* (2011). Under the influence of radiation pressure, particles' trajectories can depart significantly from Keplerian counterparts, thus modifying the exosphere structure. As a consequence, the sodium atoms might be expected to be pushed away from the Sun towards the





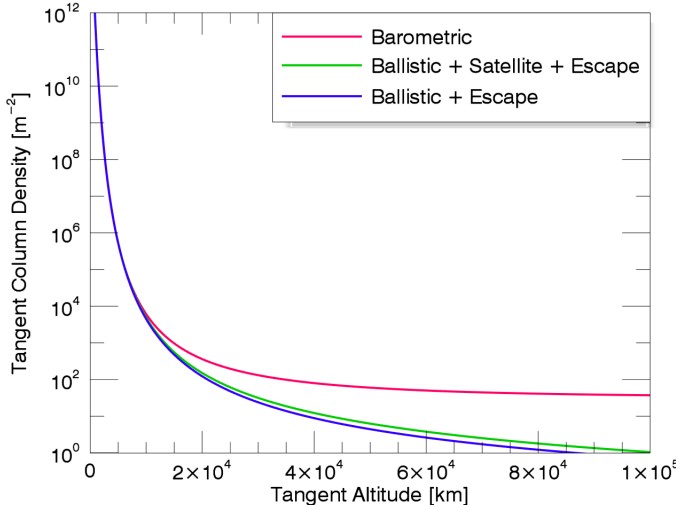

**Figure A1.** Examples of the TCD profiles for sodium for an exobase temperature of 594 K using the Chamberlain model. The pink curve represents the density profile for the barometric law, the green curve represents a combination of ballistic + satellite + escape orbits, and the blue curve represents density profiles including ballistic + escaping particles.

night-side of Mercury as the radiation pressure increases. It has been shown that for sodium atoms, radiation acceleration can be up to 54% of the surface gravity (*Smyth and Marconi*, 1995). We follow *Bishop and Chamberlain* (1989) by modifying the potential energy function, $|\lambda(r)|$, and we implement the solar radiation acceleration expression as used by *Wang and Ip* (2011). Equation A3 is the new expression for the potential energy in units of $k_B T$ and is a combination of the acceleration by gravity and radiation forces:

$$\lambda(r) = \frac{GMm}{k_B T r} - \frac{m b_{rp} r \cos(\theta)}{k_B T} \qquad (A3)$$

where $b_{rp}$ is the radiation acceleration and is a function of TAA. We used the value of $b_{rp}$ from *Smyth and Marconi* (1995) for TAA=160°. $\theta$ is the solar zenith angle. Figure A2 is another example of sodium column density profiles considering different values of surface temperature, true anomaly angle (TAA) and solar zenith angle (SZA). The variation with TAA modifies substantially the density profiles. When the radiation pressure is maximal, i.e., TAA≈ 80°, 280° and SZA=0° (subsolar point), the density profiles have a steeper slope, which means that exospheric particles are pushed back towards the surface. On the other hand, when the radiation pressure is minimal but not zero (that is when SZA=90°), i.e., TAA= 80°, 180° and SZA=89°, the density profiles have a flatter slope, which means that exospheric particles are able to reach higher altitudes.



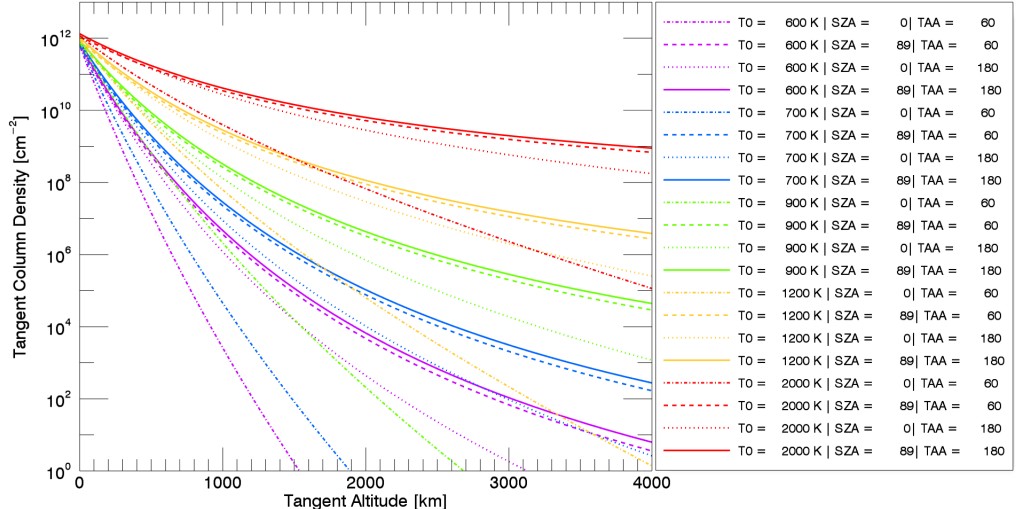

**Figure A2.** Examples of the TCD profiles for sodium using Chamberlain model for different values of temperature, $T_0 = [600, 700, 900, 1200, 2000]$ K, and fixing parameters for maximum and close to minimum radiation pressure (max: TAA $\approx 60°$ and SZA = 0°, ∼min: TAA $\approx 180°$ and SZA = Uzcanga 89°).

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
