# Peer review of "Mercury's Subsolar Sodium Exosphere: An *ab initio* Calculation to Interpret MASCS/UVVS Observations from MESSENGER"

_Annales Geophysicae, 2018_

## Referee Comment (RC1) · R. Killen (Referee) · 8 Nov 2018

Review:

Mercury's Sodium Exosphere: An ab initio Calculation to Interpret MASCS/UVS Observations from MESSENGER

Gamborino, Vorburger and Wurz Ann. Geophys. , 2018

1. This paper concludes that thermal desorption dominates all other processes in the production of sodium in Mercury's exosphere. There are several mistakes made in coming to this conclusion.

[Figure]

First, on page 14 the scale height of thermally desorbed atoms at the subsolar point was listed as 57 km. However, it was already shown by Cassidy et al. (2015), and previously by Bishop, that the scale height at the subsolar point is reduced by radiation pressure by a factor of 1/ (g + mbcos(theta)) which in this case is 40%. Thus the actual scale height is 40 km. The implication of this is that MASCS would never have seen these particles even if they were there because MASCS did not scan below 50 km.

More importantly however is the use of the full number density of Na in the crystalline lattice in this calculation. It is known that thermal desorption only acts on adsorbed atoms. As discussed by Farrrell et al. (2015) an atom on the surface of a space-weathered planet will only execute a few oscillations before finding and becoming trapped in a deep potential well. Farrell et al. conclude based on observations of H and OH at the Moon that: "We point out that diffusion times of H migrating outward also apply to H migrating inward, deeper into the regolith. We have not investigated this possibility, but presume that the H trapped in a vacancy (high U) cannot easily migrate outward to space or inward to deeper locations. It is effectively trapped." This conclusion must apply to all species, not just H. "It is more likely that the loitering H retention is very mild (1% per lunation), and when it gets too large is offset by other loss processes like impact vaporization and sputtering."

W. M. Farrell, D.M. Hurley, M.I. Zimmerman, Solar wind implantation into lunar regolith: Hydrogen retention in a surface with defects. Icarus 255 (2015) 116–126

2. Thermal desorption: page 4 line 20:"The flux of thermally released Na atoms is given by n0vth, where vth is the mean speed." In fact the release must be integrated over the Boltzmann distribution.

3. Micro-meteorite vaporization

The reference to Borin et al, 2009 should be updated. I believe that this paper was revised and the flux was revised downward.

4. Sputtering; The reference to Collier et al. (2001) is mis-quoted. What they actually said was "Neutral particles in this energy range, which encompass most of the plasma in the heliosphere, can result when energetic particles charge exchange with the Earth's hydrogen geocorona."

Since Mercury does not have an extensive hydrogen corona with the density of the Earth's geocorona, this charge exchange is not going to happen at Mercury. The solar wind does not have a neutral component. The neutral's were measured inside the Earth's geocorona due to charge exchange.

5. Other comments

Page 1: The existence of oxygen: the Mariner 10 observations were generous upper limits. MESSENGER actually has a new limit of 2 R.

R. J. Vervack Jr., R. M. Killen, W. E. McClintock, A. W. Merkel, M. H. Burger, T. A. Cassidy, and M. Sarantos. New discoveries from MESSENGER and insights into Mercury's exosphere. Geophys. Res. Lett., 10.1002/2016GL071284

Page 2 line 4: MESSENGER also measured the sodium tail: McClintock, W. E. et al., Mercury's Exosphere: Observations During MESSENGER's First Mercury Flyby. Science 321, 92 - 94, 2008.

More recent observations were by Carl Schmidt et al.

Figure 2: The normalization of all sources to a column density of 1011 cm-2 at the surface is not realistic and is misleading.

Please also note the supplement to this comment:
https://www.ann-geophys-discuss.net/angeo-2018-109/angeo-2018-109-RC1-supplement.pdf

---

## Author Comment (AC1) · 16 Nov 2018

R. K. Comment:This paper concludes that thermal desorption dominates all other processes in the production of sodium in Mercury's exosphere.

Reply: Just as a remark, the former statement is only true for the specific day of observation, TAA, and observation geometry. Something we mention and discuss throughout the paper.

R. K. Comment: There are several mistakes made in coming to this conclusion. First, on page 14 the scale height of thermally desorbed atoms at the subsolar point was

listed as 57 km. However, it was already shown by Cassidy et al. (2015), and previously by Bishop, that the scale height at the subsolar point is reduced by radiation pressure by a factor of 1/ (g + mbcos(q)) which in this case is 40%. Thus the actual scale height is 40 km. The implication of this is that MASCS would never have seen these particles even if they were there because MASCS did not scan below 50 km.

Reply: Following is a table showing the different Scale Height values, H, for T=594K, m=22.99amu, TAA=158°(Rorb=0.458AU), subsolar point, g=3.703 m/s2, gr=mbcos(sza)=0.453 m/s2 (Smyth, 1986):

– Theoretical (from barometric formula, no radiation pressure): H = kBT / mg = 58 km – Theoretical + radiation pressure: H = kBT / m(g+gr) = 51.6 km – Our numerical model: simulating TD + radiation pressure, and g=g(h) Altitude, h, at which: TD_density_data(h=0) / e = 57 km

In our numerical calculation, the scale height is calculated from the density profile and we consider the variation of g with altitude (something not considered in the barometric formula). We look at when density is reduced by 1/e and this agrees with the Chamberlain theory. Both, Monte Carlo and Chamberlain theory have full implementation of the photon pressure at the given TAA of the observation. Both MC and Chamberlain match the observations. We are confident that these results are right. We agree that at a couple of scale heights the signal is much lower than at the surface but this thermal signal is observed by MASCS.

R. K. Comment: More importantly however is the use of the full number density of Na in the crystalline lattice in this calculation. It is known that thermal desorption only acts on adsorbed atoms.

Reply: Indeed, thermal desorption acts only on adsorbed atoms, an assumption that we make from the beginning (see section 3.3). The full number density of Na is only used to simulate the "source" population (produced by SP and MIV), whereas for the "ambient" population (produced by TD and PSD) the number density is calculated from

the resulting returning flux from SP and MIV. We revised the text to make sure this is clear (see second paragraph in section 4).

R. K. Comment: As discussed by Farrrell et al. (2015) an atom on the surface of a space weathered planet will only execute a few oscillations before finding and becoming trapped in a deep potential well. They conclude that: "We point out that diffusion times of H migrating outward also apply to H migrating inward, deeper into the regolith. We have not investigated this possibility, but presume that the H trapped in a vacancy (high U) cannot easily migrate outward to space or inward to deeper locations. It is effectively trapped." This conclusion must apply to all species, not just H. "It is more likely that the loitering H retention is very mild (1% per lunation), and when it gets too large is offset by other loss processes like impact vaporization and sputtering." W. M. Farrell, D.M. Hurley, M.I. Zimmerman, Solar wind implantation into lunar regolith: Hydrogen retention in a surface with defects. Icarus 255 (2015) 116–126

Reply: We agree that the processes of interactions of atoms on realistic regolith surface are complicated and the energetics of adsorptions of the atoms on the regolith grains are varied. Hydrogen atoms are chemically very reactive species, actually are radicals, and thus will behave differently compared to metallic atoms, thus generalization from H to Na cannot be made straight forward. The observations presented in Cassidy et al. (2015) of the near surface Na exosphere are nicely reproduced by the MC and the Chamberlain model using thermal desorption in a quantitative way.

R. K. Comment: Thermal desorption: page 4 line 20:"The flux of thermally released Na atoms is given by n 0 vt h , where v th is the mean speed." In fact the release must be integrated over the Boltzmann distribution.

Reply: We agree and have revised the text and corrected the mistake. The theoretical thermal flux is indeed proportional to the integral of the Maxwell-Boltzmann distribution. However, as explained in Sections 3.3 and 5.1, we actually calculate this flux as the sum of the returning flux from MIV, SP, and the diffusion-limited exospheric flux (see

expression 8 in Section 5.1).

R. K. Comment: Micro-meteorite vaporization: The reference to Borin et al, 2009 should be updated. I believe that this paper was revised and the flux was revised downward.

Reply: Thanks for your comment. Indeed, in Borin et al. (2010) the flux was reduced by a factor of ∼2.6, still high. The reference Borin et al. (2009) is only used by us to give and idea of the range of uncertainty, but we actually use the values given by Müller et al. (2002).

R. K. Comment: Sputtering; The reference to Collier et al. (2001) is mis-quoted. What they actually said was "Neutral particles in this energy range, which encompass most of the plasma in the heliosphere, can result when energetic particles charge exchange with the Earth's hydrogen geocorona." Since Mercury does not have an extensive hydrogen corona with the density of the Earth's geocorona, this charge exchange is not going to happen at Mercury. The solar wind does not have a neutral component. The neutral's were measured inside the earth's geocorona due to charge exchange.

Reply: Thanks for pointing that out. We put the wrong reference there. In the Collier et al. (2003) paper it is shown that there is a neutral solar wind component that originates from the solar wind – dust interaction near the Sun. We have added the right reference.

R. K. Comment: Other comments: Page 1: The existence of oxygen: the Mariner 10 observations were generous upper limits. MESSENGER actually has a new limit of 2 R. R. J. Vervack Jr., R. M. Killen, W. E. McClintock, A. W. Merkel, M. H. Burger, T. A. Cassidy, and M. Sarantos. New discoveries from MESSENGER and insights into Mercury's exosphere. Geophys. Res. Lett., 10.1002/2016GL071284

Reply: Thanks for your comment. This reference was added to the text (see Introduction).

R. K. Comment: Page 2 line 4: MESSENGER also measured the sodium tail: Mc-

Clintock, W. E. et al., Mercury's Exosphere: Observations During MESSENGER's First Mercury Flyby. Science 321, 92 - 94, 2008. More recent observations were by Carl Schmidt et al.

Reply: Thanks for your comment. Reference was added to the text in that same line.

R. K. Comment: Figure 2: The normalization of all sources to a column density of 10 11 cm -2 at the surface is not realistic and is misleading.

Reply: Thanks for your remark. We agree that the normalization does not make sense and the main purpose of this figure is to show the different shapes and slopes of the tangent altitude profiles when varying the release mechanisms and characteristic temperatures. To avoid confusion, we have normalized the tangent column density at the surface to one and re-did Figure 2.

Please also note the supplement to this comment:
https://www.ann-geophys-discuss.net/angeo-2018-109/angeo-2018-109-AC1-supplement.pdf

**Supplement:**

**Rosemary Killen's Review:**
Mercury's Sodium Exosphere: An ab initio Calculation to Interpret MASCS/UVS Observations from MESSENGER

Gamborino, Vorburger and Wurz
Ann. Geophys. , 2018

1. This paper concludes that thermal desorption dominates all other processes in the production of sodium in Mercury's exosphere.
Reply: Just as a remark, the former statement is only true for the specific day of observation, TAA, and observation geometry. Something we mention and discuss throughout the paper.

There are several mistakes made in coming to this conclusion.
First, on page 14 the scale height of thermally desorbed atoms at the subsolar point was listed as 57 km. However, it was already shown by Cassidy et al. (2015), and previously by Bishop, that the scale height at the subsolar point is reduced by radiation pressure by a factor of $1/(g + mbcos(q))$ which in this case is 40%. Thus the actual scale height is 40 km. The implication of this is that MASCS would never have seen these particles even if they were there because MASCS did not scan below 50 km.
Reply: Following is a table showing the different Scale Height values, H, for T=594K, m=22.99amu, TAA=158°($R_{orb}$=0.458AU), subsolar point, g=3.703 m/s$^2$, $g_r$=mbcos(sza)=0.453 m/s$^2$ (Smyth, 1986):

| Theoretical (from barometric formula, no radiation pressure) | $H = k_BT / mg$ | 58 km |
|---|---|---|
| Theoretical + radiation pressure | $H = k_BT / m(g+g_r)$ | 51.6 km |
| Our numerical model: simulating TD + radiation pressure, and g=g(h) | Altitude, h, at which: TD_density_data(h=0) / e | 57 km |

In our numerical calculation, the scale height is calculated from the density profile and we consider the variation of g with altitude (something not considered in the barometric formula). We look at when density is reduced by 1/e and this agrees with the Chamberlain theory. Both, Monte Carlo and Chamberlain theory have full implementation of the photon pressure at the given TAA of the observation. Both MC and Chamberlain match the observations. We are confident that these results are right. We agree that at a couple of scale heights the signal is much lower than at the surface but this thermal signal is observed by MASCS.

More importantly however is the use of the full number density of Na in the crystalline lattice in this calculation. It is known that thermal desorption only acts on adsorbed atoms.
Reply: Indeed, thermal desorption acts only on adsorbed atoms, an assumption that we make from the beginning (see section 3.3). The full number density of Na is **only** used to simulate the "source" population (produced by SP and MIV), whereas for the "ambient" population (produced by TD and PSD) the number density is calculated from the resulting returning flux from SP and MIV. We revised the text to make sure this is clear (see second paragraph in section 4).

As discussed by Farrrell et al. (2015) an atom on the surface of a space weathered planet will only execute a few oscillations before finding and becoming trapped in a deep potential well. They conclude
that: "We point out that diffusion times of H migrating outward also apply to H migrating inward, deeper into the regolith. We have not investigated this possibility, but presume that the H trapped in a vacancy (high U) cannot easily migrate outward to space or inward to deeper locations. It is effectively trapped." This conclusion must apply to all species, not just H.
"It is more likely that the loitering H retention is very mild (1% per lunation), and when it gets too large is offset by other loss processes like impact vaporization and sputtering."
W. M. Farrell, D.M. Hurley, M.I. Zimmerman, Solar wind implantation into lunar regolith: Hydrogen retention in a surface with defects. Icarus 255 (2015) 116–126

Reply: We agree that the processes of interactions of atoms on realistic regolith surface are complicated and  the energetics of adsorptions of the atoms on the regolith grains are varied. Hydrogen atoms are chemically very reactive species, actually are radicals, and thus will behave differently compared to metallic atoms, thus generalization from H to Na cannot be made straight forward. The observations presented in Cassidy et al. (2015) of the near surface Na exosphere are nicely reproduced by the MC and the Chamberlain model using thermal desorption in a quantitative way.

2. Thermal desorption: page 4 line 20:"The flux of thermally released Na atoms is given by n 0 vt h , where v th is the mean speed." In fact the release must be integrated over the Boltzmann distribution.
Reply: We agree and have revised the text and corrected the mistake. The theoretical thermal flux is indeed proportional to the integral of the Maxwell-Boltzmann distribution. However, as explained in Sections 3.3 and 5.1, we actually calculate this flux as the sum of the returning flux from MIV, SP, and the diffusion-limited exospheric flux (see expression 8 in Section 5.1).

3. Micro-meteorite vaporization: The reference to Borin et al, 2009 should be updated. I believe that this paper was revised and the flux was revised downward.
Reply: Thanks for your comment. Indeed, in Borin et al. (2010) the flux was reduced by a factor of ~2.6, still high. The reference Borin et al. (2009) is only used by us to give and idea of the range of uncertainty, but we actually use the values given by Müller et al. (2002).

4. Sputtering; The reference to Collier et al. (2001) is mis-quoted. What they actually said was "Neutral particles in this energy range, which encompass most of the plasma in the heliosphere, can result when energetic particles charge exchange with the Earth's hydrogen geocorona." Since Mercury does not have an extensive hydrogen corona with the density of the Earth's geocorona, this charge exchange is not going to happen at Mercury. The solar wind does not have a neutral component. The neutral's were measured inside the earth's geocorona due to charge exchange.
Reply: Thanks for pointing that out. We put the wrong reference there. In the Collier et al. (2003) paper it is shown that there is a neutral solar wind component that originates from the solar wind – dust interaction near the Sun. We have added the right reference.

5. Other comments:
Page 1: The existence of oxygen: the Mariner 10 observations were generous upper limits. MESSENGER actually has a new limit of 2 R.
R. J. Vervack Jr., R. M. Killen, W. E. McClintock, A. W. Merkel, M. H. Burger, T. A. Cassidy, and M. Sarantos. New discoveries from MESSENGER and insights into Mercury's exosphere. Geophys. Res. Lett., 10.1002/2016GL071284

Reply: Thanks for your comment. This reference was added to the text (see introduction).

Page 2 line 4: MESSENGER also measured the sodium tail: McClintock, W. E. et al., Mercury's Exosphere: Observations During MESSENGER's First Mercury Flyby. Science 321, 92 - 94, 2008. More recent observations were by Carl Schmidt et al.
Reply: Thanks for your comment. Reference was added to the text in that same line.

Figure 2: The normalization of all sources to a column density of 10 11 cm -2 at the surface is not realistic and is misleading.
Reply: Thanks for your remark. We agree that the normalization does not make sense and the main purpose of this figure is to show the different shapes and slopes of the tangent altitude profiles when varying the release mechanisms and characteristic temperatures. To avoid confusion, we have normalized the tangent column density at the surface to one and re-did Figure 2.

[revised manuscript text omitted]

---

## Referee Comment (RC2) · Milillo (Referee) · 4 Jan 2019

General Comments This paper reports the results of the MC simulation of Na exosphere at a specific position along the Mercury's orbit and compare it to the MESSENGER /MASCS observations. The authors conclude that close to the surface the Na atoms released by thermal desorption are the main constituents, whereas the main mechanisms able to transport Na at higher altitudes is the micro-meteorite impact vaporization. The results are interesting and original, and also the summary figures at the end are a nice schematization. Nevertheless there are some lacks in the explanations and in the description. The model is specifically computed at TAA 160°, that is,

quite close to apohelion (low radiation pressure), and it is limited to equatorial region for comparing it to the MESSENGER observations. This is not clear in the title, in the abstract and in the first part of the paper, while it is an important point since different release mechanisms can act at different surface regions (local time, and latitudes). For instance, the title should be "Mercury's subsolar Sodium exosphere:..." Generally the paper does not consider adequately the recent relevant literature on the subject, especially in the introduction. I invite the author to update the introduction with more recent and relevant papers. Detailed comments are reported here below.

Specific comments: page 1 line18: Oxygen is not the main issue here, anyway, if the authors want to mention it, the Mariner 10 detection was an upper limit.

page 1 line 21: here the references are not adequate. There are many important observations from different telescopes, especially here the observations from the THEMIS solar telescope and from the McMath-Pierce telescope cannot be neglected.

page 1 line 23: if "these" refers to MESSENGER, it is not true. If the ground based observations are the observations showing high latitude enhance the references again are lacking of relevant literature.

page 2 line 10: before Leblanc and Johnson, 2010, Sarantos et al 2009 for the Moon and Mura et al. 2009 for Mercury suggested that the release processes influence to each other.

page 3 line 9-11: this is a repetition

page 3 line 15: Also here the references are not adequate: the Na short scale time variability has been analyzed by Massetti et al. 2017, this reference must be included here.

page 3 line 20: this last sentence should be moved with some more discussion in the conclusions section

page 3 line 24: "amid", there is a typo.

page 4 line 3: "fr", again here there are typos,

section 3.1: it is not clear if the authors consider average conditions or TAA variability or surface position. Some clarification on possible dependence by these factors that could affect the conclusions should be given.

page 5 line 7-9: this sentence is not clear. It needs further explanations

page 5 line 9-10: please quantify the contribution of neutral component. The heavy ions components are also relevant especially during CME (Kallio et al. 2008) since the yield is much higher than for protons or neutral hydrogen

page 6 line 25: I would not write that ion fluxes onto the surface and yields are low. Is it low with respect to what?

page 6 line 30: I would write that the MIV contribution is estimated as comparable to SP (in fact it is not known)

page 7 line 15: the figures should be numbered in sequence.

section 4: The first part is a repetition, while it should be stated clearly in the text that the model is applied to a specific TAA and SZA, as is listed in the Table 1. So the result applies to this specific situation.

page 8 line 9: the figures should be numbered in sequence.

Figure 4 and 5 : I suggest to put these two figures together as left and right since are essentially the same, and a comparison would be easier.

page 13 lines 8 and 9: some typos

page 13 eq 6 and 7: Here I am confused, I think that not all the available "free" Na is thermally desorbed. It depends by the temperature. So a probability weight should be applied to the ambient source. I would consider that the Na in the exosphere for TD release is source for TD=(TD+PSD+MIV+SP + Diff)* Prob=TD+ambient*chi where TD,

PSD, MIV and SP are the return fluxes In fact the free Na is available also for other release processes. This complicates the discussion. Eq 6 should be true if Prob = 1 and return flux for PSD =0. These assumptions are tacit, while they are explained later. I think that the treatment should be done clearer, also because it is not valid everywhere.

page 14 line 3: delete "radial column density" here.

page 14 line 4: add "Where" before vth and move the sentence before "The radial column density..."

page 14 line 12: this sentence is not clear.

page 14 line 22-23: not clear , please explain better what is the suggested mechanism.

Figure 7 caption: correct "Taken from the results of our model"

---

## Author Comment (AC2) · 21 Jan 2019

Reply to Anna Milillo's review: Mercury's subsolar Sodium Exosphere: An ab initio Calculation to Interpret MASCS/UVS Observations from MESSENGER

Gamborino, Vorburger and Wurz Ann. Geophys., 2018

Dear Editor, We thank Dr. Anna Milillo for her comprehensive review of our manuscript. Her comments and advice helped us to improve it. We have addressed all the points mentioned and revised the manuscript accordingly. In the text below you will find the reply to each comment. Your sincerely, Diana Gamborino.

[Figure]

General Comments: This paper reports the results of the MC simulation of Na exosphere at a specific position along the Mercury's orbit and compare it to the MESSENGER /MASCS observations. The authors conclude that close to the surface the Na atoms released by thermal desorption are the main constituents, whereas the main mechanisms able to transport Na at higher altitudes is the micro-meteorite impact vaporization. The results are interesting and original, and also the summary figures at the end are a nice schematization. Nevertheless there are some lacks in the explanations and in the description. The model is specifically computed at TAA 160 âŮę , that is, quite close to apohelion (low radiation pressure), and it is limited to equatorial region for comparing it to the MESSENGER observations. This is not clear in the title, in the abstract and in the first part of the paper, while it is an important point since different release mechanisms can act at different surface regions (local time, and latitudes). For instance, the title should be "Mercury's subsolar Sodium exosphere:. . ." Generally the paper does not consider adequately the recent relevant literature on the subject, especially in the introduction. I invite the author to update the introduction with more recent and relevant papers. Detailed comments are reported here below. Reply: We have changed the title.

Specific comments: page 1 line18: Oxygen is not the main issue here, anyway, if the authors want to mention it, the Mariner 10 detection was an upper limit. Reply: We agree and we have removed this sentence.

page 1 line 21: here the references are not adequate. There are many important observations from different telescopes, especially here the observations from the THEMIS solar telescope and from the McMath-Pierce telescope cannot be neglected. Reply: We have properly added these references to the same line.

page 1 line 23: if "these" refers to MESSENGER, it is not true. If the ground based observations are the observations showing high latitude enhance the references again are lacking of relevant literature. Reply: We have corrected this in the same line.

page 2 line 10: before Leblanc and Johnson, 2010, Sarantos et al 2009 for the Moon and Mura et al. 2009 for Mercury suggested that the release processes influence to each other. Reply: We have properly added these references to the same line.

page 3 line 9-11: this is a repetition Reply: Indeed this sentence is repeated. We have removed it.

page 3 line 15: Also here the references are not adequate: the Na short scale time variability has been analyzed by Massetti et al. 2017, this reference must be included here. Reply: We have properly added these references to the same line.

page 3 line 20: this last sentence should be moved with some more discussion in the conclusions section. Reply: We have moved this sentence to the conclusions (previous to last paragraph), and elaborated on it.

page 3 line 24: "amid", there is a typo. Reply: We decided to remove the whole paragraph because is rather unnecessary.

page 4 line 3: "fr", again here there are typos, Reply: We have removed the typo.

section 3.1: it is not clear if the authors consider average conditions or TAA variability or surface position. Some clarification on possible dependence by these factors that could affect the conclusions should be given. Reply: We use the parameters from the given observation conditions, which are: TAA=158°, subsolar point (latitude=0°, noon). This parameters can be found on Table 1. A discussion regarding different observations conditions can be found in the last part of the Results and Discussion, as well as in the Conclusions.

page 5 line 7-9: this sentence is not clear. It needs further explanations Reply: Indeed, it was not clear. We have re-written this sentence.

page 5 line 9-10: please quantify the contribution of neutral component. The heavy ions components are also relevant especially during CME (Kallio et al. 2008) since the yield is much higher than for protons or neutral hydrogen Reply: We have added
the neutral component upper limit reported by Collier et al. (2003). For regular SW conditions, as prevail during the observations, only protons and alpha particles are important.

page 6 line 25: I would not write that ion fluxes onto the surface and yields are low. Is it low with respect to what? Reply: We made this sentence clearer in the same line.

page 6 line 30: I would write that the MIV contribution is estimated as comparable to SP (in fact it is not known) Reply: We agree that the MIV and SP contributions to the exosphere are comparable when solar wind SP is active. We have made this sentence clearer in the same line.

page 7 line 15: the figures should be numbered in sequence. Reply: We have changed the order of the figures. We renamed Figure 2 to Figure 3 and vice versa. Figure 3 is now at the end of the section 4 with its reference text in last paragraph of this section.

section 4: The first part is a repetition, while it should be stated clearly in the text that the model is applied to a specific TAA and SZA, as is listed in the Table 1. So the result applies to this specific situation. Reply: We have corrected, specified the specific observation conditions, and avoided repetition.

page 8 line 9: the figures should be numbered in sequence. Reply: We have corrected this.

Figure 4 and 5 : I suggest to put these two figures together as left and right since are essentially the same, and a comparison would be easier. Reply: We have put these figures together.

page 13 lines 8 and 9: some typos Reply: We have corrected the typos.

page 13 eq 6 and 7: Here I am confused, I think that not all the available "free" Na is thermally desorbed. It depends by the temperature. So a probability weight should be applied to the ambient source. I would consider that the Na in the exosphere for TD release is source for TD=(TD+PSD+MIV+SP + Diff)* Prob=TD+ambient*chi where TD,

PSD, MIV and SP are the return fluxes In fact the free Na is available also for other release processes. This complicates the discussion. Eq 6 should be true if Prob = 1 and return flux for PSD =0. These assumptions are tacit, while they are explained later. I think that the treatment should be done clearer, also because it is not valid everywhere. Reply: We agree with you that the free Na is available for other release processes and that it depends on the surface temperature, as you say. We find that for the temperature we use and observation geometry, TD is more than 10 orders of magnitude more efficient in releasing particles compared to PSD. This is explained in detail in section 5.2 where we invoke the vapor pressure calculation into the argument. We find that, even if TD would reduce its release to a factor of 100 over PSD, TD would still dominate and justify that PSD is not effective in this particular situation. This is why we ignore PSD in the returning flux. Having neglected PSD, the weighted probability factor is not included because we assume that all the Na that returns to the subsolar point will be thermally desorbed rapidly. We agree that this is not valid everywhere and not necessarily the case for other observation conditions, where PSD or other surface release mechanisms might be active and compete with each other. We have improved the text in the Conclusion section, where we dedicate a new paragraph (previous to the last) to make clear that our results apply only to the observation conditions of the data we analyzed.

page 14 line 3: delete "radial column density" here. Reply: correction made.

page 14 line 4: add "Where" before vth and move the sentence before "The radial column density..." Reply: The v_th has to be introduced after the definition of the surface density. We have added the "where" that was missing.

page 14 line 12: this sentence is not clear. Reply: We have changed the sentence and made it clearer.

page 14 line 22-23: not clear , please explain better what is the suggested mechanism. Reply: We have changed the sentence and made it clearer.

Figure 7 caption: correct "Taken from the results of our model" Reply: This has been corrected.

Please also note the supplement to this comment:
https://www.ann-geophys-discuss.net/angeo-2018-109/angeo-2018-109-AC2-supplement.pdf

<hr>

---

## Author Comment (AC3) · 28 Mar 2019

We corrected the value of the TAA in the new manuscript. We originally calculated a TAA of 158° for the radial orbital distance of 0.458 AU (day of observation: 23.04.2012). However, after using the JPL's HORIZONS Ephemeris Tool we found that the TAA is actually equal to 202°. Nevertheless, both angles correspond to the same radial orbital distance and have the same radiation pressure acceleration, thus our results do not change. The new manuscript with the TAA correction and the highlighted corrections from reviewers is enclosed as a supplement in this comment.

[Figure]

Please also note the supplement to this comment:
https://www.ann-geophys-discuss.net/angeo-2018-109/angeo-2018-109-AC3-supplement.pdf
* * *
[Figure]

**Supplement:**

[revised manuscript text omitted]

---

## Author Response (AR1)

**REPLY TO REVIEWERS**
Mercury's **subsolar** Sodium Exosphere: An ab initio Calculation to Interpret MASCS/UVS Observations from MESSENGER

Diana Gamborino, Audrey Vorburger, and PeterWurz
**Submitted to Ann. Geophys. , 2018**

Dear Editor,
We thank Dr. Rosemary M. Killen and Dr. Anna Milillo for their comprehensive reviews of our manuscript. Their comments and advice helped us to improve our work substantially.

We have addressed all the points mentioned in their discussions, and revised the manuscript accordingly. We made a few updates in the reply to the reviewers.

In the text below you will find the reply to each comment (blue and red font) and the respective reference to the corrected manuscript.

Your sincerely,
Diana Gamborino.

**(A) Reply to Rosemary M. Killen's Review:**

1. This paper concludes that thermal desorption dominates all other processes in the production of sodium in Mercury's exosphere. There are several mistakes made in coming to this conclusion.
First, on page 14 the scale height of thermally desorbed atoms at the subsolar point was listed as 57 km. However, it was already shown by Cassidy et al. (2015), and previously by Bishop, that the scale height at the subsolar point is reduced by radiation pressure by a factor of 1/ (g + mbcos(q)) which in this case is 40%. Thus the actual scale height is 40 km. The implication of this is that MASCS would never have seen these particles even if they were there because MASCS did not scan below 50 km.

We agree that at a couple of scale heights the signal is much lower than at the surface but this thermal signal is still observed by MASCS. We show now the different ways one can calculate the scale height.
Following is a table showing the different Scale Height (H) values we computed using the actual observation pararemeters: $T_s$=594K, $m_{Na}$=22.99 amu, **TAA=158°($R_{orb}$=0.458 AU)**, subsolar point → sza= 0°, g=3.703 m/s$^2$, $g_r$=**mbcos(sza)=0.453 m/s$^2$** (Smyth, 1986):

| | | |
|---|---|---|
| Theoretical (from barometric formula, **no** radiation pressure). | $H = k_B T / mg$ | 58 km |
| Theoretical + radiation pressure. | $H = k_B T / m(g+g_r)$ | 51.6 km |
| **Our numerical model:** simulating Thermal Desorption (TD) + radiation pressure, **and g=g(h)**. | Altitude, h, at which: TD_density_data(h=0) / e | 57 km |

The radiation pressure can indeed reduce the scale height by almost 50% at the subsolar point, when it reaches its maximum value but this happens only around TAA~90° and TAA~270° (Smyth, 1986). For a TAA=158° the radiation pressure is close to the minimum.

On the other hand, the scale height in our numerical calculation is calculated from the density profile, and we consider the variation of g with altitude (something that is not considered in the barometric formula). We check when density is reduced by 1/e to determine the scale height, which can vary a bit with every run of the code and ensemble size. The upper table shows that our nummerical caculation is overestimating the theoretical value of the scale height when radiation pressure is considered, and this is partly due to our implementation of g=g(h). Both, our Monte Carlo model and Chamberlain model have full implementation of the photon pressure at the given TAA of the observation and both match the observations.

More importantly however is the use of the full number density of Na in the crystalline lattice in this calculation. It is known that thermal desorption only acts on adsorbed atoms.

Indeed, thermal desorption acts only on adsorbed atoms, an assumption that we make from the beginning (see section 3.3). The full number density of Na is only used to simulate the "source" population (produced by SP and MIV), whereas for the "ambient" population (produced by TD and PSD) the number density is calculated from the resulting returning flux from SP and MIV. We revised the text to make sure this is clear (see page 9, lines 10-13).

As discussed by Farrrell et al. (2015) an atom on the surface of a space weathered planet will only execute a few oscillations before finding and becoming trapped in a deep potential well. They conclude that: "We point out that diffusion times of H migrating outward also apply to H migrating inward, deeper into the regolith. We have not investigated this possibility, but presume that the H trapped in a vacancy (high U) cannot easily migrate outward to space or inward to deeper locations. It is effectively trapped." This conclusion must apply to all species, not just H. "It is more likely that the loitering H retention is very mild (1% per lunation), and when it gets too large is offset by other loss processes like impact vaporization and sputtering." W. M. Farrell, D.M. Hurley, M.I. Zimmerman, Solar wind implantation into lunar regolith: Hydrogen retention in a surface with defects. Icarus 255 (2015) 116–126

We agree that the processes of interactions of atoms on realistic regolith surface are complicated and the energetics of adsorptions of the atoms on the regolith grains are varied. The Hydrogen atom is a chemically highly reactive species, actually it is a radical, and thus will behave very differently compared to metallic atoms. Therefore, generalization from H to Na cannot be made straight forward. The observations presented in Cassidy et al. (2015) of the near surface Na exosphere are nicely reproduced by our MC and the Chamberlain model using thermal desorption in a quantitative way, using the physical parameters during the time of the observation.

2. Thermal desorption: page 4 line 20:"The flux of thermally released Na atoms is given by n 0 vt h , where v th is the mean speed." In fact the release must be integrated over the Boltzmann distribution.

We agree and have revised the text, and corrected the mistake. The theoretical thermal flux is indeed proportional to the Maxwell-Boltzmann distribution integrated in the velocity space. However, as explained in 5.1, we actually calculate the thermal flux as the sum of the returning flux to the surface from MIV, SP, and the diffusion-limited exospheric flux (see expression 8 in Section 5.1.)

3. Micro-meteorite vaporization: The reference to Borin et al, 2009 should be updated. I believe that this paper was revised and the flux was revised downward.

Indeed, in Borin et al. (2010) the flux was reduced by a factor of ~2.6, still high. The reference Borin et al. (2009) is only used by us to give and idea of the range of uncertainty, but we actually use the values given by Müller et al. (2002). See corrections in page 4, line 28; page 5, lines 2-3.

4. Sputtering; The reference to Collier et al. (2001) is mis-quoted. What they actually said was "Neutral particles in this energy range, which encompass most of the plasma in the heliosphere, can result when energetic particles charge exchange with the Earth's hydrogen geocorona." Since Mercury does not have an extensive hydrogen corona with the density of the Earth's geocorona, this charge exchange is not going to happen at Mercury. The solar wind does not have a neutral component. The neutral's were measured inside the earth's geocorona due to charge exchange.

Thanks for pointing that out. We put the wrong reference there. In the Collier et al. (2003) paper it is shown that there is a neutral solar wind component that originates from the solar wind – dust interaction near the Sun. We have added the right reference (see page 5, line 17.)

5. Other comments:
Page 1: The existence of oxygen: the Mariner 10 observations were generous upper limits. MESSENGER actually has a new limit of 2 R. R. J. Vervack Jr., R. M. Killen, W. E. McClintock, A. W. Merkel, M. H. Burger, T. A. Cassidy, and M. Sarantos. New discoveries from MESSENGER and insights into Mercury's exosphere. Geophys. Res. Lett., 10.1002/2016GL071284

Thanks for pointing this out. However, we have decided to remove this sentence because, as Dr. Milillo mentioned, the observation of Oxygen is not the main issue of our paper.

Page 2 line 4: MESSENGER also measured the sodium tail: McClintock, W. E. et al., Mercury's Exosphere: Observations During MESSENGER's First Mercury Flyby. Science 321, 92 - 94, 2008. More recent observations were by Carl Schmidt et al.

Thanks for your comment. We have added the reference to the text (page 2, line 5.)

Figure 2: The normalization of all sources to a column density of 10 11 cm -2 at the surface is not realistic and is misleading.

Thanks for your remark. We agree that the normalization does not make sense and we have corrected this issue. Actually, the main purpose of this plot is to show the different shapes, magnitdues, and slopes of the tangent altitude profiles when we include

different release mechanisms and different characteristic temperatures. To avoid confusion, we have normalized the tangent column density at the surface to one and re-did . You will now find this plot in Figure 3.

**(B) Reply to Anna Milillo's review:**

**General Comments:** This paper reports the results of the MC simulation of Na exosphere at a specific position along the Mercury's orbit and compare it to the MESSENGER /MASCS observations. The authors conclude that close to the surface the Na atoms released by thermal desorption are the main constituents, whereas the main mechanisms able to transport Na at higher altitudes is the micro-meteorite impact vaporization. The results are interesting and original, and also the summary figures at the end are a nice schematization. Nevertheless there are some lacks in the explanations and in the description. The model is specifically computed at TAA 160 ∘ , that is, quite close to apohelion (low radiation pressure), and it is limited to equatorial region for comparing it to the MESSENGER observations. This is not clear in the title, in the abstract and in the first part of the paper, while it is an important point since different release mechanisms can act at different surface regions (local time, and latitudes). For instance, the title should be "Mercury's subsolar Sodium exosphere:. . ." Generally the paper does not consider adequately the recent relevant literature on the subject, especially in the introduction. I invite the author to update the introduction with more recent and relevant papers. Detailed comments are reported here below.

Thanks, we have changed the title to be more precise.

**Specific comments:**
page 1 line18: Oxygen is not the main issue here, anyway, if the
authors want to mention it, the Mariner 10 detection was an upper limit.

We agree and we have removed this sentence.

page 1 line 21: here the references are not adequate. There are many important
observations from different telescopes, especially here the observations from the
THEMIS solar telescope and from the McMath-Pierce telescope cannot be neglected.

We have properly added these references to the same line.

page 1 line 23: if "these" refers to MESSENGER, it is not true. If the ground based
observations are the observations showing high latitude enhance the references again
are lacking of relevant literature.

We have corrected the misleading references and adapted the text accordingly (see page 1, lines 19-24 and 25; page 2, lines 1-2.)

page 2 line 10: before Leblanc and Johnson, 2010, Sarantos et al 2009 for the Moon and Mura et al. 2009 for Mercury suggested that the release processes influence to each other.

We have properly added these references (see page 2, line11.)

page 3 line 9-11: this is a repetition

Thanks for pointing that out. Indeed, this sentence is repeated so we have removed it.

page 3 line 15: Also here the references are not adequate: the Na short scale time variability has been analyzed by Massetti et al. 2017, this reference must be included here.

We have properly added this reference (see page 3, line 12.)

page 3 line 20: this last sentence should be moved with some more discussion in the conclusions section.

We have moved this sentence to the conclusions and elaborated on it (page 17, lines 12-15.)

page 3 line 24: "amid", there is a typo.

We decided to remove the whole paragraph because is rather unnecessary.

page 4 line 3: "fr", again here there are typos,

We have removed the typo.

section 3.1: it is not clear if the authors consider average conditions or TAA variability or surface position. Some clarification on possible dependence by these factors that could affect the conclusions should be given.

We use the parameters from the given observation conditions, which are: TAA=158°, subsolar point (latitude=0°, noon). This parameters can be found on Table 1. A discussion regarding different observations conditions can be found in the last part of the Results and Discussion, as well as in the Conclusions (see page15, lines 5-11;  page 17, lines 12-15.)

page 5 line 7-9: this sentence is not clear. It needs further explanations

Indeed, it was not clear. We have re-written this sentence (see page 5, lines 13-16.)

page 5 line 9-10: please quantify the contribution of neutral component. The heavy ions components are also relevant especially during CME (Kallio et al. 2008) since the yield is much higher than for protons or neutral hydrogen

We have added the neutral component upper limit reported by Collier et al. (2003) (page 5, line 17). For regular SW conditions, as prevail during the observations, only protons and alpha particles are important.

page 6 line 25: I would not write that ion fluxes onto the surface and yields are low. Is it low with respect to what?

Thansk for pointing this out, we made this sentence clearer (page 6, line 31; page 7, line 1.)

page 6 line 30: I would write that the MIV contribution is estimated as comparable to SP (in fact it is not known).

We agree that the MIV and SP contributions to the exosphere are comparable when solar wind SP is active. We have made this sentence clearer (page 7, line 6-8.)

page 7 line 15: the figures should be numbered in sequence.

We have changed the order of the figures. We renamed Figure 2 to Figure 3 and vice versa. Figure 3 is now at the end of Section 4.

section 4: The first part is a repetition, while it should be stated clearly in the text that the model is applied to a specific TAA and SZA, as is listed in the Table 1. So the result applies to this specific situation.

Thanks for pointing this out. We have removed the first pararagraph in Section 4, and specified the observation conditions.

page 8 line 9: the figures should be numbered in sequence.

Thanks, we have corrected this.

Figure 4 and 5 : I suggest to put these two figures together as left and right since are essentially the same, and a comparison would be easier.

We have put these figures together and changed the text accordingly (page 10, lines 29-30; page 11, lines 1-9.)

page 13 lines 8 and 9: some typos

Thanks, we have corrected the typos.

page 13 eq 6 and 7: Here I am confused, I think that not all the available "free" Na is thermally desorbed. It depends by the temperature. So a probability weight should be applied to the ambient source. I would consider that the Na in the exosphere for TD release is source for TD=(TD+PSD+MIV+SP + Diff)* Prob=TD+ambient*chi where TD, PSD, MIV and SP are the return fluxes In fact the free Na is available also for other release processes. This complicates the discussion. Eq 6 should be true if Prob = 1 and return flux for PSD =0. These assumptions are tacit, while they are explained later. I think that the treatment should be done clearer, also because it is not valid everywhere.

We agree with you that the free Na is available for other release processes and that it depends on the surface temperature, as you say. We find that for the temperature we use and observation geometry, TD is more than 10 orders of magnitude more efficient in releasing particles compared to PSD. This is explained in detail in section 5.2 where we invoke the vapor pressure calculation into the argument. We find that, even if TD would reduce its release to a factor of 100 over PSD, TD would still dominate and justify that PSD is not effective in this particular situation. This is why we ignore PSD in the returning flux. Having neglected PSD, the weighted probability factor is not included because we assume that all the Na that returns to the subsolar point will be thermally desorbed rapidly.

We agree that this is not valid everywhere and not necessarily the case for other observation conditions, where PSD or other surface release mechanisms might be active and compete with each other.
We have improved the text in the Conclusion section, where we dedicate a new paragraph (see page 17, lines 12-15.) to make clear that our results apply only to the observation conditions of the data we analyzed.

page 14 line 3: delete "radial column density" here.

Thanks for pointing that out. Correction made.

page 14 line 4: add "Where" before vth and move the sentence before "The radial column density..."

Thanks, we have corrected this (page 13, line 28.)

page 14 line 12: this sentence is not clear.

We have changed the sentence and made it clearer (page 14, lines 6-7.)

page 14 line 22-23: not clear , please explain better what is the suggested mechanism.

We have changed the sentence and made it clearer (page 15, lines 5-7.)

Figure 7 caption: correct "Taken from the results of our model"

Thanks, we have corrected this. You will find this correction in the caption now of Figure 6.

[revised manuscript text omitted]

---

## Referee Report (RR1)

Second Review of:

**Mercury's Subsolar Sodium Exosphere: An ab initio Calculation to Interpret MASCS/UVVS Observations from MESSENGER**

Diana Gamborino, Audrey Vorburger, and Peter Wurz

1. The response to the referee comments are adequate except for the discussion of scale height. Actually what the authors say is correct for the true anomaly angles they calculate. However, the data that they are using, which is from Cassidy et al. (2015), are at TAA angles between 65 and 70 degrees, where the radiation pressure is maximum. In this case the radiation pressure is 50% of gravitational pressure, and the scale height is reduced to 2/3 of the no radiation pressure value. Therefore the scale height for T=594 is 39 km and the scale height for 1200 K (appropriate for PSD) is 78 km. I believe that the fit to the data is actually shallower than what is shown in the Gamborino paper, and fits the 78 km scale height quite well. This was the conclusion of Cassidy et al. (2015) and is why Cassidy et al. conclude that the data are consistent with photon-stimulated desorption.

Also note that the maximum radiation pressure is at 65° not 90° as stated by the author because of the combined influence of heliocentric distance and radial velocity caused by the ellipticity of the orbit. If the simulated scale height is 57 km as they state, then the scale height that would have been attained without radiation pressure would have been 85.5 km, consistent with a temperature of 875 K. This is too hot to be consistent with thermal vaporization.

2. I still have additional questions about the results on thermal desorption.

In order to review the results in the Gamborino paper I asked the question, what is the thermal desorption rate at the subsolar point of Mercury?

These are my calculations:

1. The adsorption time is given by

$$\tau_{ads} = \frac{1}{v}\exp(\frac{Q}{kT})$$

where $v = 10^{13}$ s$^{-1}$
Q= 2 eV
T= 594 K

Then $\tau_{ads}=10^{-13} \exp(3.2\times10^{-12}$ erg$/(1.38\times10^{-16} * 594)$

$\tau_{ads}=8.99\times10^{3}$ sec

The surface number density is given by

$n_{surf} = (dn/dt)\tau_{ads}$

Assume that 1/3 of the upward impact vaporization flux is lost. Then the downward flux is 2/3 Fup

The surface number density is then $2/3\ F_{up} * 9 \times 10^3$

$n_{surf} = 8.4 \times 10^{10}$ Na cm$^{-2}$

This is the *adsorbed* number density on the surface, not in the exosphere.

What is the thermal desorption rate, $K_{des}$?

$K_{des} = A_{des}\ exp(- E_{des}/kT)$

A = vibration frequency = $10^{13}$ s$^{-1}$

E=0.8 - 2.35 eV

If E=2 eV then the desorption rate (per atom) is

$K_{des} = 1.1 \times 10^{-4}$ s$^{-1}$

The thermal desorption rate is then:

$\phi_{therm} = n_{surf} * K_{des} = 9.3 \times 10^6$ cm$^{-2}$ s$^{-1}$

 This means that the thermal desorption rate is equal to the impact vaporization rate. It turns out that this is independent of the adsorption time.

The ratio of the scale heights is
$H_{IV}/H_{therm} = 6.7$

the ratio of the number densities is

$n_{0therm}/n_{0IV} = 2.6$

If the fluxes are the same, the surface density in the exosphere at the surface has to be in the ratio of the square root of the temperatures (i.e. outward velocity). That means that there are 2.6 times as many thermal atoms at the surface than IV atoms but the scale height of the thermal atoms is 0.15 that of the IV atoms. The thermal atoms would not be seen.

3. Compare Figure 4 from Gamborino et al. with the above calculation. If the surface number density ratio of thermal to IV is 2.6 what would be the ratio of surface tangent columns?

The tangent column is proportional to the surface number density and the square root of the scale height.
The ratio of the surface number densities is 2.6, and the square root of the ratios of the scale heights is 0.385, so that the product is exactly 1.00.
*That means that the surface tangent columns of the thermal sodium and the IV are the same. The thermal component would not be seen in this case. It is curious that the same result is obtained independently of the adsorption energy because the number density at the surface is inversely related to the desorption rate, limiting the thermal component.*

*The thermal to hot surface tangent columns in Gamborino et al. Figure 4 shows a ratio of about 250.* If the adsorbed atoms are derived from the primary source (section 3.3) (e.g. Impact vaporization) then this cannot be the case. Other possibilities are that the PSD source partially thermalizes on impact with the surface (I think that Smyth came to this conclusion a long time ago.) But the other conclusion (from statement 1 above) is that the scale height of the cold component is more consistent with PSD, not thermal desorption.

4. Page 13: The number density (in the exosphere) at the surface given on Gamborino line 29 is the same as I estimated for the adsorbed surface number density (which is not the same thing).

5. It is not possible to understand the ratios of column densities in Table 2. I derived that the ratio of surface exospheric number density of thermal to IV is 2.6. The ratio of thermal to IV scale height is about 0.13, which we both agree. Therefore the ratio of the thermal column to IV column should be 0.34. Given that, the IV column would be $2x10^{10}$ cm$^{-2}$, or two orders of magnitude less than that given in table 2, column 3.

6. The other questions I have are that the column abundance given for PSD in Table 2 is greater than the maximum possible for a collisionless exosphere, and that given for SP is larger than the observed abundance. These serious questions that need to be addressed because Mercury has an exosphere, not an atmosphere.

Rosemary Killen
Referee

---

## Referee Report (RR2)

Review of Gamborino et al., Mercury's Sodium Exosphere

**Section 3.3 line 12:**

"These particles are thermally accommodated to the local surface temperature..."

From Yakshinskiy and Madey, Surface Science 593, 202-209, 2005:

Because the mass of Na is smaller than
Si and larger than O in the $SiO_2$ film, a quasi-elastic
binary collision can lead to backscattering of
incoming Na from Si , but not from O; in comparison,
more massive K would not backscatter from
either O or Si. Potassium has a higher probability
than Na of losing energy to substrate phonons,
and being trapped.

I read this to mean that Na can backscatter without losing energy to the substrate, thus not accommodating to the surface temperature.

Figure 1:

Why are you using a temperature of 594 K for PSD. Again I show the measured velocity distribution of Na from PSD.

[Figure]

"The desorbing Na is suprathermal (~900 K) with respect to a 100 K substrate )Yakshinskiy and Madey, Icarus 168, 53-59, 2004).

[Figure]

**Figure 2** Velocity distribution for ESD of neutral Na from SiO$_2$ (0.22 ML); electron energy $E_e = 200$ eV. The electron source is pulsed, and Na atoms that desorb from SiO$_2$ have flight times of ~50 μs to the detector Ir ribbons (see Fig. 1 legend). The time-of-flight technique is highly sensitive for detection of Na atoms, with low background signal. dN is the number of desorbing atoms having velocities within the range $V + dV$.

Velocity distribution for substrate temperature of 250 K (Yakshinskiy and Madey, Nature Lett, 1999.)

There is clearly a surface temperature dependence but the PSD temperature is about 800 K higher than the surface temperature.

Also a temperature of 4000 K for MIV is higher than usually inferred but not completely out of line.

Section 5.
lines 13 - 14: "Using the Chamberlain theory implies that the only way to increase the praticle's characteristic temperature (and thus able to reach high altitudes) is by increasing the surface temperature."

I disagree with this statement. You simply assume a source temperature. The Chamberlain exosphere is only going to give you an estimate of what is going on with a surface-bounded exosphere but you can assume any source temperature you want. The source can be IV or PSD or whatever. Also if you are using MC as well as Chamberlain you can assume any value of sticking and any value of thermal accommodation between 0 and 1. Thus the MC model does not have to rely on a thermal source at the surface temperature or any given temperature or any velocity distribution. According to your paper you have a MC model.

Line 1 page 13: "The Chamberlain model works fine only for an exospheric population in thermal equilibrium with the surface temperature."

This is incorrect. Actually it works fine as long as the atoms do not exchange energy with the surface. According to Yakshinskiy and Madey they do not thermalize to the surface temperature, so keeping the different sources separate works fine.

Page 13 line 8: "The PSD TCD profile does not fit quite as well to the observations and it has to be scaled with a factor of $4\times10^{-4}$ to match part of the tail."

Given that you used the same temperature for PSD as thermal desorption I don't see that you have two different populations.

line 9: delete Uzcanga.

Line 14: "Moreover the TCD from SP falls off much less with altitude than the observations."

Perhaps you should change the binding energy in the equation.

Equation 6: I think that each source needs to be scaled by the source flux times the time per ballistic hop times the number of bounces per unit time.
In other words, the ambient source rate times the lifetime per bounce times the number of bounces equals the sum of the source terms times their respective bounce time times the number of bounces in the same time ...

It is not clear whether you have done this.

line 27: losses assumed are ionization and gravitational escape. What about loss to high energy activation sites on the surface?

6. Conclusions:

The conclusions of this paper are that TD dominates governed by a surface temperature of 594 K.

It needs to be made clear that the velocity distribution assumed for PSD was the same as for thermal vaporization and is much less than the measured temperature for PSD. Therefore no PSD was used in this model. That needs to be explicitly stated. No conclusions can be reached about whether there is PSD or not if PSD was not included in the model. Similarly sputtering is probably not an important source at the equator where these data were taken. This needs to be clearly stated that this is not a global model.  The temperature assumed for IV is probably too high to be realistic and a lower temperature IV plus a small amount of sputter would probably match just as well.

---

## Author Response (AR2)

**REPLY to the**
Second Review of:
Mercury's Subsolar Sodium Exosphere: An ab initio Calculation to
Interpret MASCS/UVVS Observations from MESSENGER
Diana Gamborino, Audrey Vorburger, and Peter Wurz

**1.** The response to the referee comments are adequate except for the discussion of scale height. Actually what the authors say is correct for the true anomaly angles they calculate. However, the data that they are using, which is from Cassidy et al. (2015), are at TAA angles between 65 and 70 degrees, where the radiation pressure is maximum. In this case the radiation pressure is 50% of gravitational pressure, and the scale height is reduced to 2/3 of the no radiation pressure value. Therefore the scale height for T=594 is 39 km and the scale height for 1200 K (appropriate for PSD) is 78 km. I believe that the fit to the data is actually shallower than what is shown in the Gamborino paper, and fits the 78 km scale height quite well. This was the conclusion of Cassidy et al. (2015) and is why Cassidy et al. conclude that the data are consistent with photon-stimulated desorption. Also note that the maximum radiation pressure is at 65° not 90° because of the combined influence of heliocentric distance and radial velocity caused by the ellipticity of the orbit. If the simulated scale height is 57 km as they state, then the scale height that would have been attained without radiation pressure would have been 85.5 km, consistent with a temperature of 875 K. This is too hot to be consistent with thermal vaporization.

Indeed, the radiation pressure acceleration is maximum for TAAs around 65°-70° and 290°-295° (see Illustration 1 taken from Smyth, 1986).
However, the derived tangent column density profile (that we use in our work) reported by Cassidy et al. (2015), corresponds to a **TAA=202°,** according to the **JPL's HORIZONS Ephemeris[3]**.

The TAA=158° that we derived is incorrect but has the same radiation pressure acceleration value because both TAAs correspond to points in the orbit with the same distance to the Sun. Therefore our results do not change but we have corrected this mistake in the manuscript. We explain in the following how we derived this angle.

The True Anomaly Angle, denoted by $v$, is derived from the definition for the distance from the Sun to Mercury, $r_{orb}$, given by the basic elliptical relation[1]:

$$r_{orb} = \frac{a(1-e^2)}{1+ecos(v)} \qquad (1)$$

therefore the TAA is equal to (in degrees):

$$v = acos\left[\frac{1}{e}\left(\frac{a(1-e^2)}{r_{orb}}-1\right)\right]\frac{180°}{\pi} \qquad (2)$$

where $e$ is the eccentricity of Mercury **(=0.20563), a** is the semimajor-axis of Mercury's orbit **(=0.387AU).** The tangent column density profiles derived by Cassidy et al. (2015) correspond to the MESSENGER observations taken on **23.04.2012** (see Illustration 3). According to the Geocentric Ephemeris of Mercury[2], on 23 April 2012 Mercury was at a distance to the Sun of **$r_{orb}$=0.458 AU**.

NATURE VOL. 323 23 OCTOBER 1986 — LETTER:

**Fig. 1** Solar radiation pressure of sodium atoms near Mercury. The radiation acceleration ($b$) experienced by sodium atoms in Mercury's atmosphere because of resonance scattering in the solar D-lines is shown in absolute and surface gravity units as a function of the true anomaly angle $f$ for the planet on its elliptical orbit around the Sun.

*Illustration 1: Radiation Pressure Acceleration as a function of True Anomaly Angle (TAA) (Smyth, 1986).*

[Figure]

*Illustration 2: Diagram of an ellipse illustrating the True Anomaly Angle. The angles 158° and 202° correspond to the same $r_{orb}$.*

If we use these values: $r_{orb}$, $e$, and $a$ in Eq. (2) we obtain: $v=158°$. However, the cosine is an even-valued function, meaning that it is symmetric around π, therefore Eq. (2) has two solutions: $v_1=158°$ and $v_2=202°$. **We used the JPLs HORIZONS System Ephemeris Tool[3] and we found that the real TAA is actually 202°. Nevertheless, note that our results do not change because the radiation pressure acceleration has the same value for TAA=158° and TAA=202°, i.e., they correspond to the same $r_{orb}$ (see Illustration 1 and 2).**

*T.A. Cassidy et al./Icarus 248 (2015) 547–559*

[Figure]

**Fig. 7.** Examples of fits including two temperature components. The individual components are shown by dashed blue lines, the sum of the two components by a solid blue line, and the data are represented by crosses. Unlike the cooler component of the exosphere, the hot component is well fit by a wide range of temperatures. In this example the hot component is equally well fit by 5000 K (left panel), 10,000 K (middle panel), and 20,000 K (right panel). For this reason we were unable to constrain the temperature of the hot component. The data were taken above the subsolar point on 23 April 2012. (For interpretation of the references to color in this figure legend, the reader is referred to the web version of this article.)

*Illustration 3: Tangent column density profiles derived from MESSENGER observations during 23.04.2012 (see caption of Fig. 7 of Cassidy et al. (2015).)*

Following Smyth (1986), we calculated that the radiation pressure acceleration is close to the minimum, for a TAA equal to 202°, and it is approximately equal to 28 cm s$^{-1}$ (see Illustration 1). This is the value we use in our calculations.

Based on this radiation pressure acceleration we get a scale height for the thermal component of Na of 57 km for a surface temperature of 594 K.

References:

**[1]** Eq. (1) and (2) for the True anomaly angle (TAA) can be found in almost any book of Orbital Mechanics,  http://spiff.rit.edu/classes/phys440/lectures/ellipse/ellipse.html

[2] $R_{orb}$=0.458 AU → distance from Mercury to the Sun on **23.04.2012** taken from the Geocentric Ephemeris for Mercury: 2012 (http://astropixels.com/ephemeris/planets/mercury2012.html)

[3] JPLs HORIZONS System: https://ssd.jpl.nasa.gov/horizons.cgi#results

**2.** I still have additional questions about the results on thermal desorption.
In order to review the results in the Gamborino paper I asked the question, what is the thermal desorption rate at the subsolar point of Mercury?
These are my calculations:
1. The adsorption time is given by

$$\tau_{ads} = (1/v) \exp(Q/kT)$$

where $v = 10^{13}$ s$^{-1}$, Q= 2 eV, T= 594 K.

An activation energy of 2 eV is way too high for a Na atom falling onto the rock surface and becoming physisorbed. Realistic energies are around 0.5 eV; 2 eV would be for a Na in a mineral compound, but it cannot be expected that a Na atom falling onto the rock forms a mineral in this process, it basically just freezes out on the surface. The vibration frequency «v» in the above calculation by the reviewer is just a generic value for zero-order estimates, not specific to a Na atom on a rock surface.

Then $\tau_{ads}$ =10$^{-13}$ exp(3.2x10$^{-12}$ erg/(1.38x10$^{-16}$ * 594)

$$\tau_{ads} = 8.99x10^3 \text{ sec}$$

The surface number density is given by $n_{surf}$= (dn/dt) $\tau_{ads}$

Assume that 1/3 of the upward impact vaporization flux is lost. Then the downward flux is 2/3 $F_{up}$

This is a coarse estimate, we use the return fluxes calculated within our model based on first principles.

The surface number density is then 2/3 $F_{up}$ * 9x10$^3$
$n_{surf}$ = 8.4x10$^{10}$ Na cm$^{-2}$

This is the adsorbed number density on the surface, not in the exosphere.
What is the thermal desorption rate, $K_{des}$ ?
$$K_{des} = A_{des} \exp(- E_{des} /kT)$$

This formula is a coarse approximation of the sublimation or evaporation from a surface. For sublimation it is better to use established measurements, e.g. Lide, D. R.: CRC Handbook of Chemistry and Physics, 84th Edition. CRC Press. Boca Raton, Florida; Section 4, 2003.

A = vibration frequency = 10$^{13}$ s$^{-1}$

This value is an estimate, but there is no support that it is the correct value for Na adsorbed on a regolith surface.

E=0.8 - 2.35 eV

If E=2 eV then the desorption rate (per atom) is $K_{des} = 1.1 \times 10^{-4}$ s$^{-1}$

The value of 2 eV corresponds to a strong binding of Na within a crystallographic compound. Usually, this value is used for sputtering of Na. Adsorbed Na, that has fallen back to the surface, will have a lower binding energy, since it is only physisorbed. Exploring the range of binding energies given above gives

$E_{des} = 0.8$ eV $\rightarrow$ $K_{des} = 1.6 \times 10^{+6}$ s$^{-1}$
$E_{des} = 2$ eV $\rightarrow$ $K_{des} = 1.1 \times 10^{-4}$ s$^{-1}$
$E_{des} = 2.35$ eV $\rightarrow$ $K_{des} = 1.1 \times 10^{-7}$ s$^{-1}$

With $E_{des} = 0.8$ eV the likely values for adsorbed Na on a surface, perhaps with even lower energy.

The thermal desorption rate is then:

$$\Phi_{therm} = n_{surf} * K_{des} = 9.3 \times 10^6 \text{ cm}^{-2} \text{ s}^{-1}$$

**$E_{des} = 0.8$ eV $\rightarrow$ $K_{des} = 1.6 \times 10^{+6}$ s$^{-1}$ $\rightarrow$ $\Phi_{therm} = 1.4 \times 10^{17}$ cm$^{-2}$ s$^{-1}$**
**$E_{des} = 2$ eV $\rightarrow$ $K_{des} = 1.1 \times 10^{-4}$ s$^{-1}$ $\rightarrow$ $\Phi_{therm} = 9.3 \times 10^{6}$ cm$^{-2}$ s$^{-1}$**
**$E_{des} = 2.35$ eV $\rightarrow$ $K_{des} = 1.1 \times 10^{-7}$ s$^{-1}$ $\rightarrow$ $\Phi_{therm} = 9.5 \times 10^{3}$ cm$^{-2}$ s$^{-1}$**

Given that Edes is not very constrained in its value, and a value of 0.8 eV or lower would be much more appropriate, the estimate by the reviewer of $K_{des}$ is rather low, and a value for the desorbed flux of $\Phi_{therm} = 1.4 \times 10^{17}$ cm$^{-2}$ s$^{-1}$ is more likely.

This means that the thermal desorption rate is equal to the impact vaporization rate. It turns out that this is independent of the adsorption time.

Based on the arguments above, we disagree, and thermal desorption is a very fast process at these temperatures.

The ratio of the scale heights is

$$H_{Iv} / H_{therm} = 6.7$$

We have calculated that the scale height for TD is 57 km and for MIV is 431 km (values derived from our numerical simulations), giving a ratio of $H_{MIV}/H_{TD} = 7.56$, pretty close to the ones of the reviewer, considering that we use a different TAA (see above).

the ratio of the number densities is

$$n_{0therm} / n_{0IV} = 2.6$$

If the fluxes are the same, the surface density in the exosphere at the surface has to be in the ratio of the square root of the temperatures (i.e. outward velocity). That means that there are 2.6 times as many thermal atoms at the surface than IV atoms but the scale height of the thermal atoms is 0.15 that of the IV atoms. The thermal atoms would not be seen.

The thermal Na are seen at low altitudes, below ~500 km. There, the observed scaled height in data reported by Cassidy et al. 2015 matches our calculation exactly, both for the MC model and for the theoretical model using the Chamberlain theory, for thermal desorption from a surface at 594 K.

**3.** Compare Figure 4 from Gamborino et al. with the above calculation. If the surface number density ratio of thermal to IV is 2.6 what would be the ratio of surface tangent columns? The tangent column is proportional to the surface number density and the square root of the scale height. The ratio of the surface number densities is 2.6, and the square root of the ratios of the scale heights is 0.385, so that the product is exactly 1.00. That means that the surface tangent columns of the thermal sodium and the IV are the same. The thermal component would not be seen in this case. It is curious that the same result is obtained independently of the adsorption energy because the number density at the surface is inversely related to the desorption rate, limiting the thermal component.  The thermal to hot surface tangent columns in Gamborino et al. Figure 4 shows a ratio of about 250. If the adsorbed atoms are derived from the primary source (section 3.3) (e.g. Impact vaporization) then this cannot be the case. Other possibilities are that the PSD source partially thermalizes on impact with the surface (I think that Smyth came to this conclusion a long time ago.) But the other conclusion (from statement 1 above) is that the scale height of the cold component is more consistent with PSD, not thermal desorption.

We calculate the scale heights for TD, MIC, and PSD from the density profiles, which result from first principles calculations, with all the details given in the manuscript. The resulting numbers are given in Table 2. From the radial density profiles we calculate the profiles for the tangential column densities. These data are given in Figure 4, and the agreement between our calculations and the observations is fine.

We agree with the referee that Na atoms falling back to the surface will thermalize upon impact on the surface, perhaps only after a few impacts. This is true for Na released by PSD, MIC, and SP.

**4.** Page 13: The number density (in the exosphere) at the surface given on Gamborino line 29 is the same as I estimated for the adsorbed surface number density (which is not the same thing).

The referee gives an adsorbed surface density as $n_{surf} = 8.4 \times 10^{10}$ Na **cm$^{-2}$**, e.g. the number of Na atoms residing at the surface. The surface density we give in Table 2 is the number density in the exosphere at the surface. We agree with the referee that these two quantities are different.

**5.** It is not possible to understand the ratios of column densities in Table 2. I derived that the ratio of surface exospheric number density of thermal to IV is 2.6. The ratio of thermal to IV scale height is about 0.13, which we both agree. Therefore the ratio of the thermal column to IV column should be 0.34. Given that, the IV column would be $2 \times 10^{10}$ cm$^{-2}$, or two orders of magnitude less than that given in table 2, column 3.

As detailed above, we disagree with the calculation of the thermal densities by the referee. However, we do want to stress that we calculate the release of Na for each release process from first principles. MIV release is calculated from the flux of impacting micro-meteorites. Thermal release is calculated from the evaporation of the ambient Na population on the surface, which is limited by the amount of available ambient Na. These processes are related to each other in that the returning Na feeds the ambient Na population at the surface.

**6.** The other questions I have are that the column abundance given for PSD in Table 2 is greater than the maximum possible for a collisionless exosphere, and that given for SP is larger than the observed abundance. These serious questions that need to be addressed because Mercury has an exosphere, not an atmosphere.

We calculate the scale height for PSD as h_PSD = 232e3 m (see Table 2).

The mean-free path at the surface for the PSD contribution can be estimated as

Lambda = 1 / (n * sigma) = 1 / (3.45e10 m^-3 * 1e-19 m^2) ≈ 2.9e8 m

Thus, h_PSD << lambda, and we are still well in the collissionless regime at the surface, and above.

Moreover, we do mention that the values for PSD are upper limits, because of the competition between TD and PSD for the ambient Na population, which is strongly in favour for TD at these temperatures. Likely, the PSD contribution is actually much smaller, we estimate by a factor > 2500. The same applies for SP, which we conclude is not relevant for the given observation parameters, i.e., observations near the sub-solar point which is shielded from the solar wind plasma. Our results show that PSD and SP for the observed parameters can not physically explain the properties of the exosphere observed, therefore we conclude that these processes are not dominant. Note that in our conclusions we mention that only TD and MIV are the best candidates to explain the observed column density profiles. No fitting was performed during our analysis.

Rosemary Killen
Referee

[revised manuscript text omitted]

---

## Author Response (AR3)

**Reply to the 3rd**
Review of Gamborino et al., Mercury's Sodium Exosphere
by Rosemary Killen

Dear Editor,
The following is the reply to the third review. Our answers are written in blue.

We thank the reviewer for her comments and have carefully elaborated on each answer.

Best regards,
Diana Gamborino and Peter Wurz

Section 3.3 line 12:
"These particles are thermally accommodated to the local surface temperature..."

From Yakshinskiy and Madey, Surface Science 593, 202-209, 2005:
"Because the mass of Na is smaller than Si and larger than O in the SiO 2 film, a quasi-elastic binary collision can lead to backscattering of incoming Na from Si , but not from O; in comparison, more massive K would not backscatter from either O or Si. Potassium has a higher probability than Na of losing energy to substrate phonons, and being trapped."

I read this to mean that Na can backscatter without losing energy to the substrate, thus not accommodating to the surface temperature.

[Figure]

Figure 1

Backscatter from surfaces without energy loss is physically impossible, the simplest approximation used in surface is the binary collision approximation (BCA) which assumes two hard spheres interact with each other. Based on BCA, having a Na atom of mass 23 interaction with a Si atom of the surface of mass 28, one would expect a significant energy exchange given the similarity of mass. The situation is not that different for Na hitting an O atom. More sophisticated models of the particle-surface interaction invoke a more realistic potential of the surface atoms, which modifies the energy exchange between the impaction atom and the surface atoms. There is no way to do a completely elastic backscattering, and this has not been observed experimentally. Yakshinskiy and Madey say "quasi-elastic".

Why are you using a temperature of 594 K for PSD. Again I show the measured velocity distribution of Na from PSD.

"The desorbing Na is suprathermal (~900 K) with respect to a 100 K substrate (Yakshinskiy and Madey, Icarus 168, 53-59, 2004).

[Figure]

**Figure 2** Velocity distribution for ESD of neutral Na from SiO$_2$ (0.22 ML); electron energy $E_e = 200$ eV. The electron source is pulsed, and Na atoms that desorb from SiO$_2$ have flight times of ~50 μs to the detector Ir ribbons (see Fig. 1 legend). The time-of-flight technique is highly sensitive for detection of Na atoms, with low background signal. dN is the number of desorbing atoms having velocities within the range $V + dV$.

Figure 2.

As explained in Gamborino and Wurz, 2018, we use a non-thermal distribution for PSD that has the actual surface temperature as one of the defining parameters.

Velocity distribution for substrate temperature of 250 K (Yakshinskiy and Madey, Nature Lett, 1999.)

There is clearly a surface temperature dependence but the PSD temperature is about 800 K higher than the surface temperature.

We fully agree with that, and the dependence on the surface temperature is fully accounted for in the used velocity distribution of PSD, see Gamborino & Wurz (2018).

In our previous work (Gamborino & Wurz, 2018) we presented a comprehensive statistical study of exactly the velocity distributions functions (VDFs) shown by the reviewer in Fig. 1 and Fig 2., which are reported by Yakshinskiy and Madey (1999, 2004). In that work we demonstrated that: "[..] *the Maxwell-Boltzmann distribution is neither statistically nor physically adequate to describe non-thermal processes such as ESD and PSD"*, if one wants to model a thermal distribution, one would need to arbitrarily choose a very high temperature to fit the observed VDFs, like Yakshinskiy and Madey later did, but with no physical interpretation.

Also a temperature of 4000 K for MIV is higher than usually inferred but not completely out of line.

We show in Figure 4 a range of temperatures for modelling MIV and find that the agreement with a temperature of 4000 K is satisfactory, perhaps a temperature somewhere between 3000 K and 4000 K would be possible.

Section 5.
lines 13 - 14: "Using the Chamberlain theory implies that the only way to increase the particles' characteristic temperature (and thus able to reach high altitudes) is by increasing the surface temperature."

I disagree with this statement. Simply a source temperature. The Chamberlain exosphere is only going to give you an estimate of what is going on with a surface-bounded exosphere but you can, (assume any source temperature you want. The source can be IV or PSD or whatever. Also if you are using MC as well as Chamberlain you can assume any value of sticking and any value of thermal accommodation 0 and 1. Thus the MC model does not have to rely on a thermal source at the surface temperature or any given temperature or any velocity distribution. According to your paper you have a MC model.

We strongly disagree with this point, and a lot of research and experiments (see references in Introduction, see Gamborino & Wurz, 2018) support that different release mechanisms produce different energy/velocity distributions of the particles released from a surface/exobase.

On the other hand, the core of the MC model is to repeatedly and randomly sample the different energy distributions associated with the different release mechanisms to obtain a numerical solution (in our case we want to determine the density vs. altitude). In fact, any Monte Carlo model relies solely on the given probability distributions.

Line 1 page 13: "The Chamberlain model works fine only for an exospheric population in thermal equilibrium with the surface temperature."

This is incorrect. Actually it works fine as long as the atoms do not exchange energy with the surface. According to Yakshinskiy and Madey they do not thermalize to the surface temperature, so keeping the different sources separate works fine.

We have to disagree with the reviewer, quoting Madey et al. JGR 1998: *A low-energy component characteristic of the surface temperature may arise from scattering of "hot" atoms in the surface regions of the porous regolith, or by thermal desorption.*

Chamberlain's theory is clear about the source of exospheric particles, stating that *"[...] the controlling factors [of exospheric particles] are gravitational attraction and **thermal energy conducted from below**"* (Chamberlain, PSS, 1963). In that same work, they also clearly state that below the exobase, *"[...] collisions maintain a complete Maxwellian distribution of velocites"*.

Furthermore, PSD is not a thermal process but a photon of sufficient energy that breaks the binding of an atom or molecule with the surface, i.e., an electronic process. Therefore PSD should not be treated like a thermal

process. Chamberlain theory assumes a Maxwell-Boltzmann distribution at the exobase, i.e., it works only for thermally desorbed particles, not for PSD. Therefore, one should not use this theory to describe atoms released via PSD.

Page 13 line 8: "The PSD TCD profile does not fit quite as well to the observations and it has to be scaled with a factor of 4x10 -4 to match part of the tail." Given that you used the same temperature for PSD as thermal desorption I don't see that you have two different populations.

We use the same surface temperature for PSD and for TD but we **do not** use the same velocity distribution for both mechanisms, therefore they **do** produce different populations. You can see that in Figure 4: TD population is the dashed-black curve, and PSD population is the vertical-dashed-grey curve, both for the same temperature.

line 9: delete Uzcanga.

This typo appears only in the first manuscript version submitted on 20.09.2018. We kindly ask the reviewer to read the latest manuscript version with corrections (submitted 29.04.2019). This typo is not there anymore.

Line 14: "Moreover the TCD from SP falls off much less with altitude than the observations."

Perhaps you should change the binding energy in the equation.

Given that observations correspond to the subsolar point, we expect Sputtering to be inactive and not an important source. Moreover, the surface binding energy is not a free parameter to be used as a parameter to fit the data, it is a material constant and thus given.

Equation 6: I think that each source needs to be scaled by the source flux times the time per ballistic hop times the number of bounces per unit time.
In other words, the ambient source rate times the lifetime per bounce times the number of bounces equals the sum of the source terms times their respective bounce time times the number of bounces in the same time...

It is not clear whether you have done this.

As described in section 3, our Monte Carlo model considers that once a particle falls back to the surface, it will be thermalized and no longer considered in the numerical integration of that particular trajectory. In case of Na, this particle becomes part of the ambient population, which subsequently will be released by thermal desorption. That is explained in detail in the manuscript.

line 27: losses assumed are ionization and gravitational escape. What about loss to high energy activation sites on the surface?

It is not clear to us what the reviewer means by "high energy activation sites", and a literature research did not provide a suitable publication. What is clear

is that the thermal desorption from the surface is driven by thermodynamics, and thus the energies available to a desorbed atom or molecule are defined by a Maxwell-Boltzmann velocity distribution.

6. Conclusions:
The conclusions of this paper are that TD dominates governed by a surface temperature of 594 K. It needs to be made clear that the velocity distribution assumed for PSD was the same as for thermal vaporization and is much less than the measured temperature for PSD. Therefore no PSD was used in this model. That needs to be explicitly stated. No conclusions can be reached about whether there is PSD or not if PSD was not included in the model. Similarly sputtering is probably not an important source at the equator where these data were taken. This needs to be clearly stated that this is not a global model. The temperature assumed for IV is probably too high to be realistic and a lower temperature IV plus a small amount of sputter would probably match just as well.

**Firstly, we do not use the same velocity distribution for TD and for PSD, and this is clearly stated in the manuscript. Please see Eq. (3) in the manuscript, this is a Weibull distribution, which is a non-thermal, long-tailed distribution adapted to the PSD process (for details see Gamborino & Wurz, 2018). For TD we used the thermal Maxwell-Boltzmann distribution, as stated in section 3.1.1. These distributions are not the same.**

**Secondly, we do include PSD in the model. Equation (3) is the velocity distribution used in our MC model and the results for TCD are shown as the vertical-dashed-grey curve in Figure 4.**

**Thirdly, we do state that SP is not an important source at the equator of Mercury. Perhaps this was not clear in the first version of the manuscript, but it is clear in the last manuscript version (please see version submitted 29.04.2019). In this version, you can see that we have stated clearly several times that SP is not important source near the equator, see the following places: page 7, line 7; page 12, lines 2–5; page 13, lines 9–13.**

Finally, Figure 4 shows how well MIV modelled by 4000 K fits the observations, and also it shows a range of temperatures to understand if there is a better temperature for modelling MIV. If there is, it would be not far below the 4000 K we used in our calculations.